# Entropy of the Multi-Channel EEG Recordings Identifies the Distributed Signatures of Negative, Neutral and Positive Affect in Whole-Brain Variability

**DOI:** 10.3390/e21121228

**Published:** 2019-12-16

**Authors:** Soheil Keshmiri, Masahiro Shiomi, Hiroshi Ishiguro

**Affiliations:** 1Advanced Telecommunications Research Institute International (ATR), Kyoto 619-0237, Japan; m-shiomi@atr.jp (M.S.); ishiguro@sys.es.osaka-u.ac.jp (H.I.); 2Graduate School of Engineering Science, Osaka University, Osaka 565-0871, Japan

**Keywords:** entropy, differential entropy, affect, brain variability

## Abstract

Individuals’ ability to express their subjective experiences in terms of such attributes as pleasant/unpleasant or positive/negative feelings forms a fundamental property of their affect and emotion. However, neuroscientific findings on the underlying neural substrates of the affect appear to be inconclusive with some reporting the presence of distinct and independent brain systems and others identifying flexible and distributed brain regions. A common theme among these studies is the focus on the change in brain activation. As a result, they do not take into account the findings that indicate the brain activation and its information content does not necessarily modulate and that the stimuli with equivalent sensory and behavioural processing demands may not necessarily result in differential brain activation. In this article, we take a different stance on the analysis of the differential effect of the negative, neutral and positive affect on the brain functioning in which we look into the whole-brain variability: that is the change in the brain information processing measured in multiple distributed regions. For this purpose, we compute the entropy of individuals’ muti-channel EEG recordings who watched movie clips with differing affect. Our results suggest that the whole-brain variability significantly differentiates between the negative, neutral and positive affect. They also indicate that although some brain regions contribute more to such differences, it is the whole-brain variational pattern that results in their significantly above chance level prediction. These results imply that although the underlying brain substrates for negative, neutral and positive affect exhibit quantitatively differing degrees of variability, their differences are rather subtly encoded in the whole-brain variational patterns that are distributed across its entire activity.

## 1. Introduction

Decoding of the brain activity that produces our conceptual knowledge is a driving force in neuroscientific research [1,2]. These efforts have led the field to realization of distinct patterns of brain activity that underlie such broad subjective experiences as individuals’ cognitive states [3] and visual imagery during sleep [4]. In this respect, affect or the individuals’ ability to experience un/pleasant feelings lays the foundation for such subjectively experienced emotions [5,6,7] and their attitudes toward them [8]. This ability that begins from the early days in individuals’ lives [9,10] also appears to be the unifying and common concept across cultures, despite their varying emphasis on the significance of emotional types [11,12].

However, the neuroscientific views of the underlying neural substrates that give rise to sensation of the affect have been broadly divided into three hypotheses—the bipolarity hypothesis [13], the bivalent hypothesis [14,15,16] and the affective workspace hypothesis [8]. Whereas the bipolarity hypothesis [13] considers the positive and negative affect to form the opposite ends of a single dimension [17,18], the bivalent hypothesis [14,15,16] emphasizes the presence of two distinct and independent brain systems for the positive and negative affect [14,15,16]. On the other hand, the affective workspace hypothesis [8] argues that the positive and negative affect are the brain states that are supported by flexible and distributed brain regions [19]. Such a divide is further escalated by the results of meta-analyses that provide contrasting yet compelling evidence for and against these hypotheses [20,21,22,23,24,25]. For instance, whereas Vytal and Hamann [20] credited the bipolarity hypothesis, Lindquist et al. [25] provided stronger support for the affective workspace hypothesis.

A common theme among these results is their focus on the change in brain activation to identify a specific [26,27,28,29] or subset [21] of the brain regions as sources of different affect. More specifically, these results do not take into account the findings that indicate that the brain activation and its information content does not necessarily modulate [30] and that the stimuli with equivalent sensory and behavioural processing demands may not necessarily result in differential brain activation [31]. In this respect, a growing body of empirical [32,33,34] and theoretical [35,36] findings provide compelling evidence for the crucial role of signal variability in the functioning of the brain. This variability that is thought to emerge from the interaction between individual neurons and neuronal circuits [37,38] occurs over broad spatiotemporal scales [39,40] and is hypothesized to reflect the cortical self-organized criticality [41,42,43,44]: a state at which cortical information capacity is maximized [45,46,47]. These observations, in turn, have motivated the viewpoints that identify the role of entropy in quantification of the variability in brain functioning [48] and cortical activity [49,50] from information processing capacity of working memory (WM) [51] and neural coding [52,53] to interplay between neural adaptation and behaviour [54], functional interactivity between the brain regions [55] and the state of consciousness [56].

In light of these findings on the importance of brain variability, we take a different approach to analysis of the neural substrates of the affect, thereby considering the whole-brain variability as it is reflected in its entropy (hereafter differential entropy (DE) [57] that is, entropy of a continuous random variable). Furthermore and unlike the previous studies, we not only consider the brain variability in terms of positive and negative affect but also contrast such a variability with the neutral state. We achieve this by utilizing the SJTU Emotion EEG Dataset (SEED) [58] that is a collection of human subjects’ sixty-two-channel EEG recordings. In this experiment, the individuals watched fifteen movie clips (four minutes in duration) whose contents elicited three distinct affect: negative, neutral and positive.

Our results suggest that the whole-brain variability significantly differentiates between the negative, neutral and positive affect. They also indicate that although some brain regions contribute more to such differences, it is the whole-brain variational pattern that results in significantly above chance level prediction of the negative, neutral and positive affect. Specifically, they identify that although the underlying brain substrates for negative, neutral and positive affect exhibit quantitatively differing degrees of variability, their differences are rather subtly encoded in the whole-brain variational patterns that are distributed across its entire activity.

## 2. Results

### 2.1. DE

Figure 1A–C plot the spatial maps of the participants’ whole-brain DE in three affect states. The higher DE in the case of negative (Figure 1A) than the neutral and the positive affect is evident in these subplots. These subplots also identify that the participants’ DE during the positive affect (Figure 1C) was higher than the neutral affect (Figure 1B). However, we observed that the difference between the negative and the positive affect was more subtle than their respective differences with the neutral affect. Figure 3A presents the grand averages of these individuals’ DEs in negative, neutral and positive affect. This subplot further verifies the participants’ differential patterns of whole-brain variability.

We found significant correlation between positive and negative (Figure 2A, left column, r = 0.67, p < 0.00001), positive and neutral (Figure 2B, left column, r = 0.87, p < 0.00001) and negative and neutral (Figure 2C, left column, r = 0.61, p < 0.00001). These correlations that were also verified by the results of their corresponding bootstrap test of significance (10,000 simulation runs) at 95.0% confidence interval (Figure 2A–C, right column) were stronger between positive and neutral. Table 1 summarizes the results of these bootstrap test of significance.

Kruskal-Wallis test identified a significant difference of the whole-brain DEs between negative, neutral and positive affect (p < 0.001, H(2, 2603) = 932.23, r = 0.60). Post-hoc Wilcoxon test (Figure 3B) also verified that the DEs associated with the negative state were significantly higher than the neutral (p < 0.001, W(1734) = 27.77, r = 0.67, M${}_{Negative}$ = 6.45, SD${}_{Negative}$ = 0.93, M${}_{Neutral}$ = 5.17, SD${}_{Neutral}$ = 0.92) and the positive (p < 0.001, W(1734) = −9.67, r = −0.23, M${}_{Positive}$ = 6.10 SD${}_{Positive}$ = 0.92) affect. It also indicated that the participants’ DEs were higher in the case of positive than neutral (p < 0.001, W(1734) = 23.29, r = 0.56).

Figure 4 shows the results of the paired two-sample bootstrap test of significance difference between DEs in negative, neutral and positive affect (10,000 simulation runs) at 95.0% (i.e., p < 0.05) confidence interval (CI). This figure confirms that the participants’ DE associated with the neutral affect was significantly lower than the positive (Figure 4A) and negative (Figure 4C) states. It also indicates that the positive DEs were significantly lower than the negative DE (Figure 4B). Table 2 summarizes these results.

### 2.2. The Linear Model

#### 2.2.1. Weights

Figure 5A shows the spatial map of the trained linear model’s weights for negative, neutral and positive affect using the whole-brain DEs. We observed that the model’s weights where within [−1⋯ 1] interval (Negative: M = 0.07, SD = 0.22, CI${}_{95.0\%}$ = [0.06 0.09], Neutral: M = −0.09, SD = 0.14, CI${}_{95.0\%}$ = [−0.10−0.08], Positive: M = 0.02, SD = 0.20., CI${}_{95.0\%}$ = [0.004 0.03]). This verified that its performance was not due to any potential overfitting. We also observed that the difference between linear model’s weights associated with the negative, neutral and positive affect were more pronounced in the frontal (FZ, F4 and F6), central (C5, C3, CZ, C4, C6), centroparietal (CP5), parietal (P5) and occipital regions (O2) (for details, see Appendix B). Specifically, the linear model appeared to characterize the negative affect (Figure 5A) with the weights that were stronger in the right frontal, centroparietal and occipital regions. Moreover, it recognized the positive affect through weights that appeared stronger (i.e., relative to the negative affect) in the left frontal as well as parietal (particularly in the left-hemisphere) regions. On the other hand, the linear model distinguished the neutral affect with the distribution of weights that were primarily negative values.

Kruskal-Wallis test indicated that the trained linear model identified these mental states with weight distributions that were significantly different (p < 0.001, H(2, 2603) = 315.98, r = 0.35). Posthoc Wilcoxon test (Figure 5B) also indicated that the model’s weights for the negative affect were larger in magnitude than the positive (p < 0.001, W(1734) = −5.56, r = 0.13) and the neutral (p < 0.001, W(1734) = 17.31, r = 0.42) affect. Weights associated with the positive affect were also significantly larger in magnitude than those associated with the neutral affect (p < 0.001, W(1734) = 11.97, r = 0.29).

#### 2.2.2. Affect Prediction

Figure 6A shows the linear model’s prediction accuracy in negative, neutral and positive affect in 1-holdout setting using whole-brain DEs. In this setting, we considered a single participant’s positive, negative and neutral affect data as test set and used the remaining participants’ data for training the linear model. We then tested the model’s performance on the holdout test data. We repeated this procedure for every participant.

Subplot (A) in Figure 6 indicates that the use of whole-brain DEs to quantify the brain variability of the participants enabled the linear model to predict their negative, neutral and positive affect with significantly above chance level (chance level accuracy ≈ 33.33%, given the three-affect settings). One-sample bootstrap test of significance (10,000 simulation runs) at 99.0% (i.e., p < 0.01) confidence interval verified this observation (M = 78.52, SD = 6.45, CI${}_{95\%}$ = [61.90 88.10]). We also observed that these predictions were substantially higher in the case of negative and neutral than the positive affect. An inspection of Figure 6A reveals that a relatively larger number of participants’ positive affect was misclassified as negative affect by the linear model. Table 3, top row, summarizes the precision, recall and F1-score associated with the linear model’s performance in negative, neutral and positive affect in 1-holdout setting using whole-brain DEs.

We further checked the linear model’s performance using the subset of channels with the similar pattern of significant differences as whole-brain DEs (Figure 6B). Our analyses (for details, see Appendix A) identified that these channels were in the frontal (F3, FZ, F4), frontocentral (FCZ), central (C5, C3, C1, C2, C4), centroparietal (CP3, CP1, CP2), parietal (P3, POZ) and occipital (CB1, OZ) regions. Figure 6B corresponds to the 1-holdout setting using this subset of participants’ channels. This subplot reveals a substantial reduction of linear model’s accuracy in comparison with the case in which we used the whole-brain DEs. Wilcoxon test identified a significant difference between the linear model’s accuracy in these two settings (p < 0.001, W(998) = 7.23, r = 0.23) which we further verified using two-sample bootstrap test of significance (10,000 simulation runs) at 99.0% (i.e., p < 0.01) confidence interval (Figure 6C) (M${}_{difference}$ = 12.54, SD${}_{difference}$ = 1.63, CI${}_{difference}$ = [8.27 16.67]) (see Appendix C for randomized 500 simulation runs of these results). Table 3, bottom row, summarizes the precision, recall and F1-score associated with the linear model’s performance in negative, neutral and positive affect in 1-holdout setting using this subset of channels with the similar pattern of significant differences as whole-brain DEs.

## 3. Materials and Methods

### 3.1. The Dataset

SEED [58] corresponds to sixty-two-channel EEG recordings (Figure 7B) of fifteen Chinese subjects (7 males and 8 females; Mean (M) = 23.27, Standard Deviation (SD) = 2.37) who watched fifteen Chinese movie clips (four minutes in duration) that elicited negative, neutral and positive affect. The authors reported that these individuals were selected based on the Eysenck Personality Questionnaire (EPQ) [59] personality traits. EPQ evaluates the individuals’ personality along three independent dimensions of temperament: Extraversion/Introversion, Neuroticism/ Stability and Psychoticism/Socialisation. Eysenck et al. [59] reported that it appears that not every individual can elicit specific emotions immediately (even in the presence of explicit stimuli) and that individuals who are extraverted and have stable moods tend to elicit the right emotions throughout the emotion-based experiments. Therefore, the authors adapted the same personality criteria that was reported by Eysenck et al. [59] to select the fifteen individuals that participated in SEED experiment.

Prior to SEED experiment, the authors asked twenty volunteers to assess a pool of movie clips in a five-point scale based on which the fifteen movie clips (i.e., five clips per negative, neutral and positive affect) whose average score were ≥ 3 and ranked in the top 5 in each affect category, were chosen. The authors further verified that the selected movie clips indeed elicited the targeted affect in a follow-up study [60] that included nine separate individuals who were different from the twenty volunteers that originally involved in rating and selection process of fifteen movie clips). The authors then used these movie clips in SEED experiment [58].

Each experiment included (Figure 7A) a total of fifteen movie clips, per participant. In this setting, each movie clip (i.e., similar to Reference [60]) was proceeded with a five-second hint to prepare the participants for its start. This was then followed by a four-minute movie clip. At the end of each movie clip, the participants were asked to answer three questions that followed the Philippot [61]. These questions were the type of emotion that the participants actually felt while watching the movie clips, whether they previously watched the original movies from which the clips were taken and whether they understood the content of those clips. The participants responded to these three questions by scoring them in the scale of 1 to 5. The participants were then instructed to take a fifteen-second rest before the next movie clip in the experiment started. Each individual participated in three experiments with an interval of about one week between them. The same set of fifteen movie clips were used in all of these three experiments. Every participant watched the same set of fifteen movie clips in the same order of their presentations.

The movie clips within each experiment were ordered in such a way that two clips with the same emotional content (e.g., both targeting negative affect) were not presented consecutively to the participants. Additionally, these clips were selected based on the criteria that their lengths were not too long to induce fatigue on the subjects while watching them, that their contents were easy to understand by the participants without any explicit explanation and that each clip elicited a single desired target affect (e.g., negative or positive).

SEED comes with its preprocessed EEG recordings which we used in the present study. Its preprocessing steps consist of downsampling the EEG recordings to 200 Hz followed by bandpass filtering the signals within 0–75 Hz. These steps were applied on the extracted EEG segments that corresponded to the duration of each movie clip. Further details on SEED experiment, EEG channels’ arrangement, movie clips’ selection criteria, data acquisition and preprocessing, labeling the emotional states associated with each movie clip and so forth, can be found in References [58,60] and http://bcmi.sjtu.edu.cn/~seed/seed.html.

### 3.2. Data Selection and Validation

In our study, we considered only the first experiment of every participant (i.e., out of their three times participations) and included all of its corresponding fifteen movie clips trails (Figure 7A). We also considered all sixty-two channels of EEG recordings of these individuals. Prior to any further analyses, we validated the selected data through the following steps.

To ensure that the DE computation was not affected by the length of EEG channels, we first ensured that all EEG recordings that were included in our study were sufficiently long in term of number of their respective data points (Mean (M) = 45,286.71, Standard Deviation (SD) = 2776.61, CI${}_{95\%}$ = [44565.70 46007.73], minimum = 37,000, maximum Length = 47,601) where CI${}_{95\%}$ refers to the 95% confidence interval. We then trimmed all participants’ EEGs to have equal lengths of 37,000 data points (i.e., the minimum length of EEG recording observed in the participants’ data).

Next, we performed Augmented Dickey Fuller (ADF) [62] and Kwiatkowski-Phillips-Schmidt-Shin (KPSS) [63] tests on all EEG channels (per participant, per affect) to ensure that they were covariance stationary and subsequently marked those participants’ data whose EEG channel(s) did not pass these tests. During these tests, we noticed that all three sessions of one of the participant did not satisfy the requirement for covariance stationarity (i.e., trend-stationarity that implies mean and variance do not change over time). Therefore, we did not include this participant in the further analyses. We also observed that two of the participants’ EEG recordings from their first sessions did not pass these tests on two of their EEG channels. Therefore, we replaced these participants’ first session EEG with their corresponding second and third sessions, respectively, that passed these tests on all of their EEG channels. Further information and insights about the mathematical background and interpretation of ADF and KPSS can be found in References [62,63,64].

### 3.3. DE Computations

We computed DE using the full-length EEG data (i.e., 37,000 data points) per channel, per participant, per affect and adapted the following procedure for its computation. For a given affect (e.g., negative), we accessed each individuals’ associated five files (i.e., one file for each of the movie clips, per affect) of 62-channel EEG recordings one-by-one. For each of these files, we then used each of the EEG channels one-by-one and computed its DE (i.e., one DE for each of the EEG channels). This resulted in five separate sets (i.e., one set for each of the movie clips) of sixty-two DEs (i.e., one DE for each of the EEG channels). Next, we utilized these five separate sets of 62 DEs and computed the average DE for each channel. For example, for channel F7 (Figure 7B), we had five DEs (i.e., one DE for each of the movie clips of a given affect (e.g., negative affect): [DE${}_{movie\;clip1}^{F7}$, DE${}_{movie\;clip2}^{F7}$, DE${}_{movie\;clip3}^{F7}$, DE${}_{movie\;clip4}^{F7}$, DE${}_{movie\;clip5}^{F7}$]. We averaged these five DEs that is, mean([DE${}_{movie\;clip1}^{F7}$, DE${}_{movie\;clip2}^{F7}$, DE${}_{movie\;clip3}^{F7}$, DE${}_{movie\;clip4}^{F7}$, DE${}_{movie\;clip5}^{F7}$]). In this respect, it is apparent that the ordering of these values have no effect on the computed average DE for F7 in a given affect state (e.g., negative affect) [65]. We repeated this procedure for each participant (i.e., fourteen) and each affect (i.e., negative, neutral and positive), thereby computing their averaged brain variability in response to a given affect. This process resulted in 1 × 62 vectors, per affect, per participant, where 62 refers to the number of EEG channels (Figure 7B).

We used a non-parametric DE estimator by Kozachenko and Leonenko [66] that estimates the differential entropy of a continuous random variable using nearest neighbour distance [67]. It is worthy of note that a number of previous studies has adapted DE for emotion classification [68,69].

### 3.4. Statistical Analysis

Our analyses included three primary steps: (1) Spearman correlation to verify the correspondence between the DEs associated with negative, neutral and positive affect states. (2) Analysis of the individuals’ whole-brain DEs in which we observed significant differences between the negative, neutral and positive affect. (3) Prediction of the negative, neutral and positive affect based on individuals’ brain variability using a linear model to determine the utility of observed significant differences in DEs for quantification of the brain variability and its functioning in response to the negative, neutral and positive affect.

#### 3.4.1. DE Analyses

First, we computed the Spearman correlations between the whole-brain DEs of every pair of affect states (i.e., positive versus neutral, positive versus negative and negative versus neutral). We followed this by computing their bootstrap (10,000 simulation runs) at 95.0% confidence intervals (i.e., p < 0.05). For the bootstrap test, we considered the null hypothesis *H0: there was no correlation between every pair of affect states’ whole-brain DEs* and tested it against the alternative hypothesis *H1: The whole-brain DEs of every pair of affect states correlated significantly*. We reported the mean, standard deviation and the 95.0% confidence interval for these tests. We also computed the *p*-value of these tests as the fraction of the distribution that was more extreme than the actually observed correlation values. For this purpose, we performed a two-tailed test in which we used the absolute values so that both the positive and the negative correlations were accounted for.

Next, we applied Kruskal-Wallis test on individuals’ whole-brain (i.e., all EEG channels) DEs, per affect, which was followed by posthoc Wilcoxon rank sum tests between every pair of affect (i.e., negative, neutral and positive). We further verified these results through application of paired two-sample bootstrap test of significance (10,000 simulation runs) at 95.0% (i.e., p < 0.05) confidence interval. For the bootstrap test, we considered the null hypothesis *H0: The difference between individuals’ whole-brain variability in two different affect was non-significant* and tested it against the alternative hypothesis *H1: The individuals’ whole-brain variability significantly differed in two different affect states*. We reported the mean, standard deviation and 95.0% confidence interval for these tests.

To further examine whether there were regions that contributed more to differences between negative, neutral and positive affect, we determined the EEG channels in which the significant difference between their DE values showed the same pattern of difference that we observed in the case of whole-brain DEs. To this end, we applied channel-wise paired two-sample bootstrap test of significance (10,000 simulation runs) at 95.0% confidence interval. For this test, we considered the null hypothesis *H0: Every channel’s DE in negative, neutral and positive affect yielded the same pattern of significant difference as in the case of whole-brain comparison* and tested it against the alternative hypothesis *H1: Only a subset of channels’ DE yielded the same pattern of significant difference as in the case of whole-brain comparison*. Subsequently, we marked the channels with such significant differences (Appendix A) and utilized them during one of the linear model’s training scenarios (Section 3.4.2). We reported the mean, standard deviation and 95.0% confidence interval for these tests.

#### 3.4.2. Linear Model Training

To determine the utility of observed significant differences in DEs for quantification of the brain variability and its functioning, we trained a linear logistic regression model to verify whether the negative, neutral and positive affect can be predicted using their corresponding DEs. For this purpose, we considered two scenarios:Use of DEs associated with all EEG channels, thereby considering the whole-brain variability. This resulted in the linear model’s input feature vectors that were of length sixty-two, per participant, per affect.Feature vectors whose DE entires were associated with the subset of EEG channels that yielded the same pattern of significant difference as in the case of whole-brain comparison (i.e., the second step in DE analyses, Appendix A).

In both scenarios, we labeled each of the above negative, neutral and positive affect input feature vectors to our training model using 0, 1 and 2 labels, respectively. We then adapted the 1-holdout setting in which we considered data associated with the negative, neutral and positive affect of a single participant as test set and used the remaining participants’ data for training the linear model. This resulted in fourteen train-test runs (i.e., first participant used as test data and the remainder of participants used for training the model, Second participant used as test data and …, fourteenth participant used as test data and the remainder of participants used for training the model). We then tested the model’s performance (i.e., per training scenario) on the holdout data. We repeated this procedure for every participant. This resulted in fourteen different test cases which we used to compute the linear model’s accuracy and confusion matrix. Given the small number of sample (i.e., fourteen participants), the total number of accuracies of the model was too small (i.e., fourteen runs) for any test of significant (parametric or non-parametric) to be meaningful. Therefore, we adapted a one-sample bootstrap test in which we tested the robustness of the model’s accuracies during the fourteen 1-holdout runs. For this test, we performed a one-sample bootstrap test of significance (10,000 simulation runs) at 99% confidence interval (CI) to check whether the model’s accuracy was above chance level (i.e., ≈33.33%). Considering the fact that we used 95% confidence interval while performing this test, its result indeed corresponded to the *p*-value *p* < 0.01 significance level.

To ensure that the observed accuracy was not due to chance, we performed a randomized model training (for both training scenarios) in which we first randomly selected an individual and then picked, at random, only one of this selected individual’s negative, neutral or positive affect data for 1-holdout testing. We then used the remaining data that did not include any of the randomly selected individual’s affect data for training. To guarantee an unbiased estimate of the linear model’s prediction, we continued this random selection until every affect was selected 500 times (Appendix C). This differed from the first 1-holdout setting in which we used data pertinent to all three affect of the randomly selected individual for testing. Given the three-affect setting in which every participant had equal number of negative, neutral and positive affect data, the chance-level accuracy was ≈33.33%.

We then compared the accuracy of these two training scenarios using Wilcoxon rank sum test. We further verified this result using two-sample bootstrap test of significance (10,000 simulation runs) at 99.0% confidence interval (i.e., p < 0.01). For this test, we considered the null hypothesis *H0: Use of whole-brain DE had no significant effect on the linear model’s prediction of the negative, neutral and positive affect* and tested it against the alternative hypothesis *H1: Use of whole-brain DE significantly improved the linear model’s prediction accuracy of the negative, neutral and positive affect*. We reported the mean, standard deviation and 99.0% confidence interval (i.e., p < 0.01) for this test.

Last, we separated the weights associated with the linear model’s training based on whole-brain DEs into three sets (i.e., one set per affect) and applied Kruskal-Wallis test on them to determine the brain regions that significantly contributed to prediction of each affect. We followed this test with posthoc Wilcoxon rank sum test between every pair of these sets. We further verified these results by applying the two-sample bootstrap test of significance (10,000 simulation runs) at 95.0% confidence interval. For the bootstrap test, we considered the null hypothesis *H0: The contribution of different brain regions to linear model’s prediction of each of the affect was non-significant* and tested it against the alternative hypothesis *H1: The brain regions contributed differentially to the linear model’s prediction of each of the affect*. We reported the mean, standard deviation and 95.0% confidence interval for these tests (Appendix B).

For the Kruskal-Wallis test, we reported the effect size $r=\sqrt{\frac{\chi^{2}}{N}}$ [70] with *N* denoting the sample size and $\chi^{2}$ is the test-statistics. In the case of Wilcoxon tests, we used $r=\frac{W}{\sqrt{N}}$ [71] as effect size with *W* denoting the Wilcoxon statistics and *N* is the sample size. All results reported were Bonferroni corrected. All analyses were carried out in Python 2.7 (Python Software Foundation, Wilmington, DE, USA) and Matlab 2016a (The MathWorks, Natick, MA, USA).

With regard to our analyses, there are two points that are worth further clarification. They are the choice of non-parametric tests and the follow-up bootstrap test of significance. Prior to our analyses, we checked the calculated DEs of the participants in each of the negative, neutral and positive affect states (separately as well as combined, with respect to the both individuals and the EEG channels for each of the affect) and found that they did not follow normal distribution. Therefore, we opted for non-parametric analyses. In the case of bootstrap, on the other hand, we realized that our analyses were performed based on a small sample of participants (i.e., fourteen individuals). Therefore, it was crucial to ensure that any significant results that we observed in our analyses were not due to a subsample of individuals’ DEs (i.e., distorted data and hence lack of central tendency). This concern was further strengthened by the result of the non-normality of DEs. Applying the bootstrap test (i.e., random sampling with replacement) that was carried out for 10,000 simulation runs (therefore allowing for potential outliers to be repeated with higher probability and more frequently) at 95% confidence interval (i.e., p < 0.05 significance level) enabled us to further verify our results.

## 4. Discussion

In this article, we sought empirical evidence for the significance of brain signal variability in quantification of the distinct affect states. For this purpose, we used SEED [58]: a publicly available collection of human subjects’ EEG recordings who watched movie clips whose contents elicited negative, neutral and positive affect. Our study was motivated by differing neuroscientific findings on the underlying neural substrates of the affect [13,14,15,16,17,18,19], thereby attributing their differences to their lack of consideration for the observations that the brain activation and its information content does not necessarily modulate [30] and that the stimuli with equivalent sensory and behavioural processing demands may not necessarily result in differential brain activation [31]. To this end, we used entropy for analysis of the whole-brain variability. Our choice of entropy for quantification of the brain signal variability was motivated by the recent findings that attribute the emergence of the brain variability to cortical self-organized criticality [41,42,43,44] that maximizes the cortical information capacity [45,46,47] for an optimal information processing [72] and therefore signify the role of entropy in quantification of the variability in brain functioning [48,49,50,51,52,53,54,55,56].

We found that the negative, neutral and positive affect were associated with differential level of brain variability. Precisely, we observed that the negative and the positive affect were associated with higher DE than the neutral state which was in accord with quantitatively higher information processing in emotional than neutral contexts [73] as well as the findings on increased brain activity with attention [74,75]. In this respect, the higher DE in the negative than the positive affect also hinted at the association between the brain activity and task difficulty [76,77] and the effect of negative emotions on cortical activity [78,79]. These results also verified the effectiveness of the brain entropy for quantification of its variational patterns in response to the negative, neutral and positive affect, in accord with the recent thesis on the entropic nature of the brain [42,43,48].

We also observed that the signatures of the negative, neutral and positive affect were present in the whole-brain variability. These results extended the previous analyses that noted such distributed emotion-specific activation patterns may provide maps of internal states that correspond to specific subjectively experienced, discrete emotions [21,25,80]. In this respect, our findings appeared to be more in line with Lindquist et al. [25] which found no single region that uniquely represented a specific affect (e.g., a region solely associated with the negative, neutral or positive), thereby implying further evidence in favour of the affective workspace hypothesis [8] than the bipolarity [13] or the bivalent hypotheses [14,15,16]. For instance, considering the bipolarity hypothesis [13] stance on attributing the positive and negative affect to the opposing ends of a single dimension [17,18], one may expect their respective brain activity to exhibit an anti-correlation. Contrary to this expectation, we observed that the brain variability associated with these affect to be positively correlated with one another. On the other hand, the emphasis by bivalent hypothesis [14,15,16] on the presence of two distinct and independent brain systems for the positive and negative affect [14,15,16] makes it plausible to expect that the observed brain activation that associates with these states to originate mostly from non-overlapping regions. Although our analyses identified a number of brain regions whose DEs significantly differed between negative, neutral and positive affect, these regions were common among these affect states. Furthermore, we also observed that limiting the linear model’s feature space to solely include these regions resulted in significantly worsening of its performance. In this regards, the affective workspace hypothesis [8] argues that the positive and negative affect are the brain states that are supported by flexible and distributed brain regions [19]. Our findings appeared to be in line with this line of reasoning due to the following observations. First, we observed that the whole-brain variability in response to negative, neutral and positive affect states were positively correlated. This indicated that any change in variational information of one affect (e.g., increase or decrease in the brain signal variability) can be explained in terms of a linear change in the variability of the other that is in the same direction as of the first one. This perspective found further evidence in the results of our simple linear model that was able to classify the negative, neutral and positive affect with a significantly above average accuracy. Second, we also observed that the variational information of these affect states were distributed in the whole-brain activity and that their differences were quantified in terms of differing level of variability in these regions’ variability (i.e., information processing as quantified by their respective DEs) than their de/activation. However, future studies and analyses are necessary to draw a more informed conclusion on these observations.

Although we noticed that a subset of channels contributed more to such distributed patterns (for details, see Appendix A), it was the whole-brain variability that yielded a significantly higher prediction power for distinguishing between the negative, neutral and positive affect. Our results suggested that the whole brain networks were more capable when compared to the brain regions that corresponded to these subset of channels. These results appeared to extend the fMRI-based findings by Saarimäki [81] that indicated that the anatomically distributed variational patterns of brain activity contained the most accurate neural signature of individuals’ mental states that underly their discrete emotions. These results were also in accord with Farroni et al. [10] and Watson and Tellegen [5] that showed the joint activity from multiple regions discriminated best between different emotions. In this respect, our results hinted at observations that the large-scale cortical networks are crucially involved in representing such high-level mental states which form the foundations for describing the distinctively elicited emotions [6,25] and that features from these cortical regions may contribute differentially during the concepts’ categorization [82].

Last, our results on the bilateral whole-brain variability that were identified through both statistical and linear model analyses appeared to be at odds with the findings that suggest a higher involvement of right-hemisphere in processing of negative affect and unpleasant emotions [83,84]. On the other hand, they were in line with the findings on the brain functioning during story comprehension that identified such bilateral activations [85,86]. This extended the results on the occurrence of such a bilateral brain activity during auditory story comprehension to the case of movies (i.e., visualized stories) as it has also been reported by Hasson et al. [87] and Jääskeläinen et al. [88].

## 5. Limitations and Future Direction

Taken together, our findings suggested the significant role of whole-brain variability in response to differential effect of the negative, neutral and positive affect. Specifically, they indicated that although the underlying brain substrates for negative, neutral and positive affect exhibited quantitatively differing degrees of variability, their differences were rather subtly encoded in the whole-brain variational patterns that were distributed across its entire activity. Further evidence for the distributed signature of the negative, neutral and positive affect in the whole-brain variability came from the significant correlation between the DEs of these affect. Interestingly, we also observed that a simple linear classifier was able to distinguish between these affect states with a significantly above chance level accuracy. Furthermore, such an ability was significantly more accurate when the linear model was trained based on the whole-brain variability. In this regard, a puzzling observation was the higher percentage of misclassification between the positive and the negative affect by the model which could not readily be attributed to the sample size since we used a balanced dataset in our analyses (i.e., equal number of samples for each of the negative, neutral and positive affect, per participant).

Another possible reason behind the observed effect could have been the difference in the shared information between these affect. However, the results of the correlation analyses (i.e., linear measure of mutual information [57]) did not provide any further insight on this matter. For instance, these results could not explain the higher misclassification of the positive than the negative affect as neutral affect by the model, despite the fact that the former shared substantially more information (i.e., substantially higher correlation) with the neutral affect than the latter did. Conversely, the lower shared information also was not sufficient to account for the observed misclassification of the positive affect since the high rate of misclassification between positive and negative should have in principle been the lowest, given their lowest correlation among the three pairwise comparisons. Additionally, we observed that the linear model misclassified the neutral affect as both positive and negative affect in an equal rate although it showed substantially different correlations with these two affect states. Although we also observed that some of the brain regions showed significant differences that was in accord with the whole-brain differential signature of the negative, neutral and positive affect, the mere use of these regions did not improve the higher percentage of the misclassification between the positive and the negative affect and only resulted in worsening of the model’s performance.

These observations implied that although the negative, neutral and positive affect appear to share a common distributed neural correlates whose signature can be identified across the whole-brain variability, such an underlying brain dynamical network might be shared differentially among them and that the negative and the positive affect might have more in common (i.e., with respect to the change in the variation in the brain information processing) than with the neutral affect. They also indicated that governing dynamics of the brain responses to these affect might not primarily be explained by linear modeling of such a common distributed neural dynamics. Therefore, future research that takes into account the non-linear dynamics of neural activity in response to negative, neutral and positive affect is necessary to further extend our understanding of the potential causes of the observed dis/similarities between these affect states.

It is also crucial to note that in the present study we utilized the entropy (i.e., DE) of EEG time series of human subjects for analysis of the whole-brain variability in response to negative, neutral and positive affect states. In this regard, research identifies a functional interactivity between the brain regions [55] in which signals from individual cortical neurons are shared across multiple areas and thus concurrently contribute to multiple functional pathways [89]. However, our analyses did not take into consideration the potential effect of such flow of information among different brain regions in response to differential affect states while computing the EEG channels’ DEs. Therefore, future research to study the possibility of such potential information flows [90,91] is necessary to better realize the utility of the brain variability in emergence of differential affect states in response to natural stimuli.

The main goal of the present study was to verify whether the underlying brain substrates for negative, neutral and positive affect were rather subtly encoded in the whole-brain variational patterns that were distributed across its entire activity. As a result, we primarily focused on the statistical analyses of the whole-brain variability in terms of its distributed information processing (i.e., its entropy) and used the linear model as a supportive evidence that showed that whole-brain variability in fact resulted in better representation of these three affect states than the use of selected brain regions that also showed similar statistically significant differences. In this respect, including several measurements of the same subjects that watched the same movie clips in the same order in multiple days would have only complicated the interpretation of the results due to such issues as multiple-comparison as well as potentially confounding factors such as redundant information (e.g., assuming no substantial change in individuals’ brain functioning, it is plausible to expect that their brain not to respond significantly differently to the same stimuli that was presented to them in different days). Therefore, we decided to reduce the possibility of occurrence of such issues and confounders by only including one out of three sessions of each of the participants. However, such multiple measurements can benefit the future research by providing an opportunity to test for the reproducibility of the current results. For instance, one can compute their respective whole-brain variability using their respective DEs and compare their corresponding neural substrates with the results in the present study. They can also be utilized as one larger test set (i.e., all together) to verify whether the linear model in this study can preserve its accuracy on this new data. The latter scenario can become a valuable testbed for the cases in which training personalized models for the individuals is desirable (e.g., personalized socially-assistive robots [92]).

Many neuroscientific studies are based on a small number participants [93,94,95,96]. For instance, the recent comprehensive meta-analysis by Lindquist [25] that reported on an extensive coverage of 914 experimental contrasts in affect-related studies accounted for 6827 participants (i.e., 6827/914 ≈ 7.47 participants on average). Furthermore, most of the individuals that are included in these studies share the same geographical and/or cultural background (but also see Reference [88] for a small deviation). Although neuropsychological findings indicate that the individuals’ ability to experience pleasant/unpleasant feelings to express these subjective mental states in terms of such attributes as positive or negative to be the unifying and common concept across cultures [11,12], future research that includes a larger human sample as well as different age groups along with more cultural diversity is necessary for drawing a more informed conclusion on the findings that were presented in this article.

## Figures and Tables

**Figure 1 entropy-21-01228-f001:**
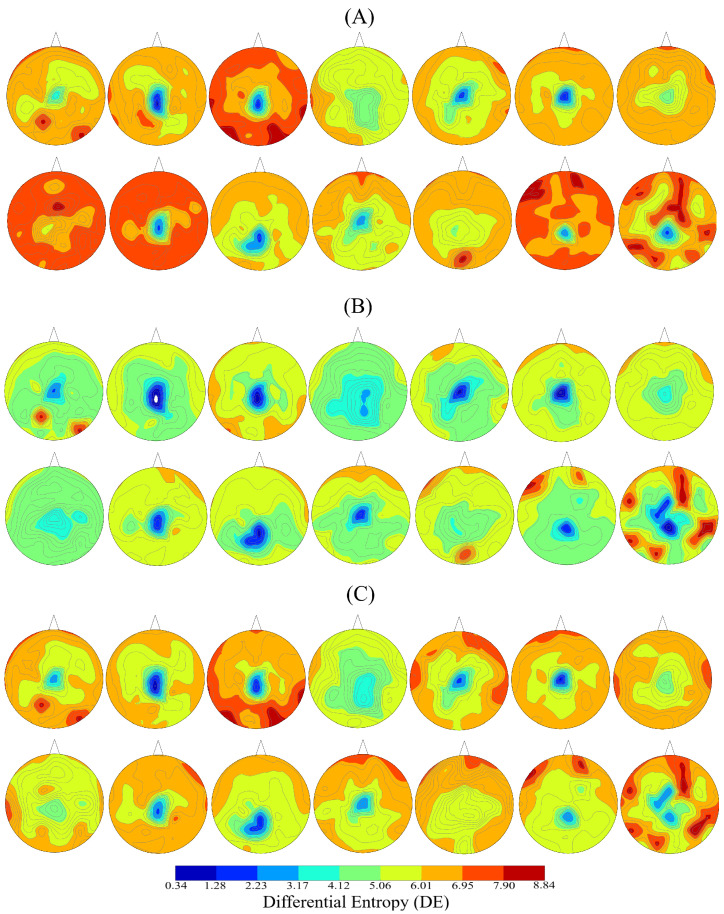
Spatial maps of participants’ whole-brain differential entropy (DE) associated with (**A**) Negative (**B**) Neutral (**C**) Positive affect. Each of these maps corresponds to one individual that was included in the present study. For each individual, we first computed (for each channel separately) the average DE (per channel) of all movie clips’ DEs that were associated with a given affect. We then used these average DEs for each channel to construct these maps. Differential patterns of participants’ whole-brain variability in three different affect states is evident in these subplots.

**Figure 2 entropy-21-01228-f002:**
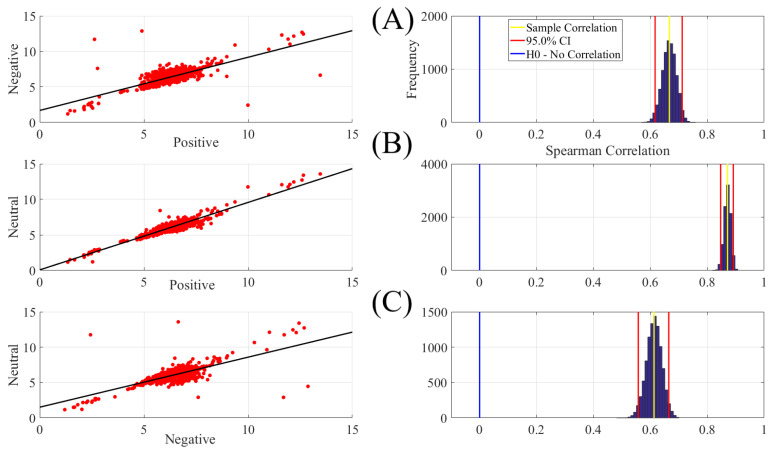
Paired Spearman correlation between participants’ DEs (**A**) positive versus negative (**B**) positive versus neutral (**C**) negative versus neutral. The subplots on the right column correspond to the bootstrap correlation test (10,000 simulation runs) at 95.0% confidence interval.

**Figure 3 entropy-21-01228-f003:**
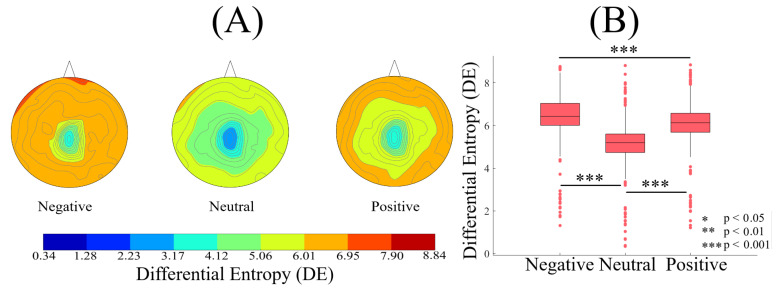
(**A**) Grand averages of the spatial maps of participants’ DEs during negative, neutral and positive affect. Differential patterns of participants’ whole-brain DE in these three different affect states is evident in these subplots. (**B**) Descriptive statistics of participants’ DEs in negative, neutral and positive affect. The asterisks mark the significant differences in these subplots.

**Figure 4 entropy-21-01228-f004:**
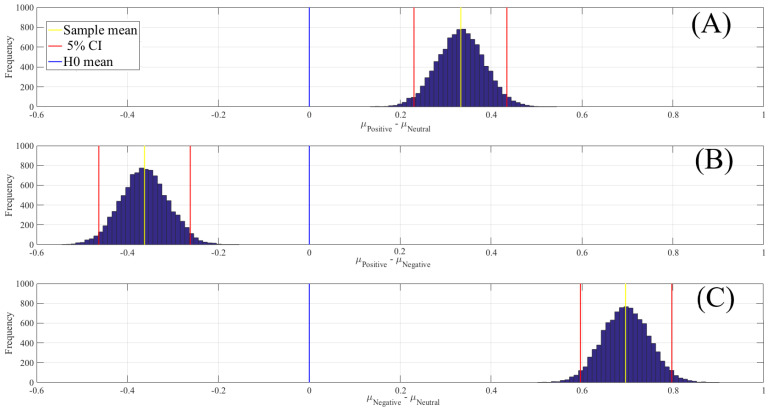
Paired two-sample bootstrap test of significance (10,000 simulation runs) at 95.0% (i.e., p < 0.05) confidence interval (CI) associated with participants’ whole-brain DEs. Compared pairs of affect are (**A**) positive versus neutral, (**B**) positive versus negative and (**C**) negative versus neutral. In these subplots, the x-axis shows $\mu_{i}-\mu_{j},\;i\neq j$ where *i* and *j* refer to one of the negative, neutral or positive affect. The blue line marks the null hypothesis H0 that is, non-significant difference between the DEs of two compared affect. The red lines are the boundaries of the 95.0% confidence interval. The yellow line shows the location of the average $\mu_{i}-\mu_{j},\;i\neq j$ for 10,000 simulation runs.

**Figure 5 entropy-21-01228-f005:**
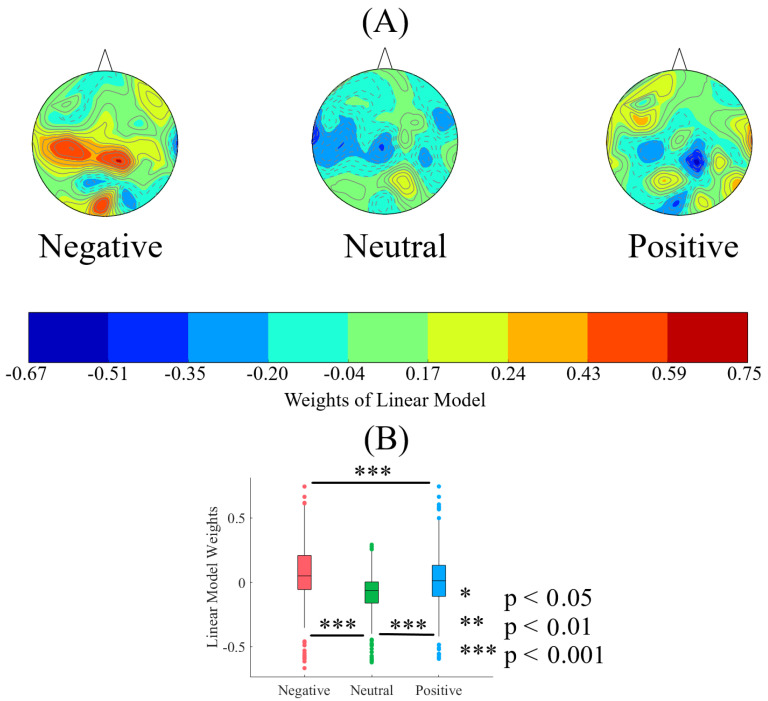
(**A**) Spatial map of weights pertinent to the trained linear model on the whole-brain DEs associated with negative, neutral and positive affect. This subplot verifies that the model’s weights where within [−1⋯ 1] interval, thereby indicating that the performance of the model was not due to any potential overfitting. Distinct pattern of model’s weights distribution associated with each of three affect is evident in these maps (see Appendix C for results on 500 randomized simulation runs). (**B**) Descriptive statistics of linear model’s weights for negative, neutral and positive affect. The asterisks mark the significant differences in these subplots.

**Figure 6 entropy-21-01228-f006:**
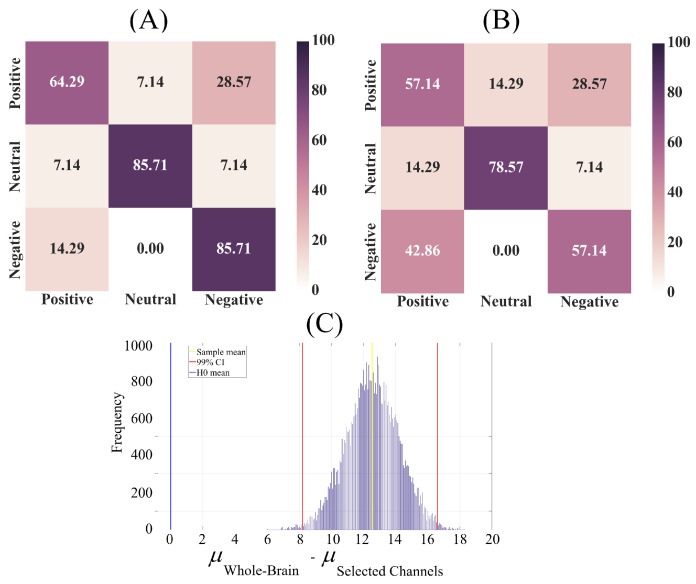
Linear model’s confusion matrices associated with its prediction accuracy in negative, neutral and positive affect in 1-holdout setting using (**A**) whole-brain DEs and (**B**) subset of channels with the similar pattern of significant differences as whole-brain DEs. These results show the model’s performance after data from every participant was used for testing. Correct predictions, per affect, are the diagonal entries of these tables and the off-diagonal entries show the percentage of each of affect that was misclassified (e.g., positive affect misclassified as negative affect). (**C**) Two-sample bootstrap test of significance (10,000 simulation runs) at 99.0% (i.e., p < 0.01) confidence interval between the accuracy of linear model using the whole-brain DEs versus subset of channels with the similar pattern of significant differences as whole-brain DEs.

**Figure 7 entropy-21-01228-f007:**
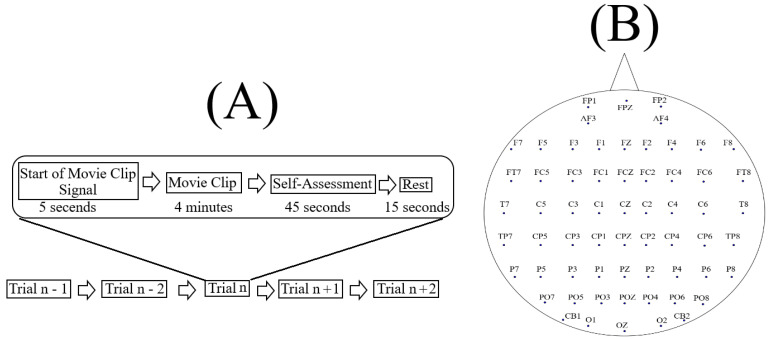
(**A**) Schematic diagram of an experiment as described in Reference [58]. Each experiment included a total of fifteen movie clips (i.e., n = 15), per participant. In this setting, each movie clip was proceeded with a five-second hint to prepare the participants for its start. This was then followed by a four-minute movie clip. At the end of each movie clip, the participants were asked to answer three questions that followed the Philippot [61]. These questions were the type of emotion that the participants actually felt while watching the movie clips, whether they watched the original movies from which the clips were taken and whether they understood the content of those clips. The participants responded to these three questions by scoring them in the scale of 1 to 5. (**B**) Arrangement of the EEG electrodes in this experiment. The sixty-two EEG channels were: FP1, FPZ, FP2, AF3, AF4, F7, F5, F3, F1, FZ, F2, F4, F6, F8, FT7, FC5, FC3, FC1, FCZ, FC2, FC4, FC6, FT8, T7, C5, C3, C1, CZ, C2, C4, C6, T8, TP7, CP5, CP3, CP1, CPZ, CP2, CP4, CP6, TP8, P7, P5, P3, P1, PZ, P2, P4, P6, P8, PO7, PO5, PO3, POZ, PO4, PO6, PO8, CB1, O1, OZ, O2, CB2.

**Table 1 entropy-21-01228-t001:** Bootstrap (10,000 simulation runs) 95.0% confidence intervals (CI) associated with the Spearman correlation between negative, Neutral and Positive affect.

Conditions	r	p (two-tailed)	CI${}_{95\%}$
Positive vs. Negative	0.67	0.00001	[0.62 0.71]
Positive vs. Neutral	0.87	0.00001	[0.85 0.89]
Negative vs. Neutral	0.61	0.00001	[0.56 0.66]

**Table 2 entropy-21-01228-t002:** Results associated with paired two-sample bootstrap test of significance (10,000 simulation runs) at 95.0% (p < 0.05) confidence interval (CI) on participants’ whole-brain DEs. Compared pairs of affect are: positive versus neutral, positive versus negative and negative versus neutral. M and SD refer to the mean difference and the standard deviation of such a difference between the two compared affect. CI shows the 95% confidence interval of the paired differences.

Conditions	M${}_{difference}$	SD${}_{difference}$	95.0% CI${}_{difference}$
Positive versus Neutral	0.33	0.05	[0.23 0.43]
Positive versus Negative	−0.36	0.05	[−0.46 −0.26]
Negative versus Neutral	0.70	0.05	[0.60 0.80]

**Table 3 entropy-21-01228-t003:** Linear model’s precision, recall and F1-score in negative, neutral and positive affect in 1-holdout setting using whole-brain DEs and subset of channels with the similar pattern of significant differences as whole-brain DEs.

Setting	Precision	Recall	F1-Score
Whole-Brain	0.86	0.71	0.77
Selected Channels	0.57	0.62	0.59

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
