# Peer review of "Entropy of the Multi-Channel EEG Recordings Identifies the Distributed Signatures of Negative, Neutral and Positive Affect in Whole-Brain Variability"

_entropy, 2019, doi:10.3390/e21121228_

Round 1

Reviewer 1 Report

The language needs to be improved. It is not clear in the abstract what is the contribution compared to the background they have mentioned mainly because it is not clear what you mean by whole-brain variability. 

It is better to start with methodology and shows the stand of your work before presenting your result. The only part that is needed to be mentioned before method section is what is used as motivation. 

Adding some results on related methods would be helpful. 

Author Response

First and foremost, the authors would like to take this opportunity to thank the reviewers and the associate editor for their time and kind consideration to review our manuscript. The comments by the reviewers enabled us to improve the quality of our results and their presentation substantially.

In what follows, we provide point-by-point responses to the comments and concerns raised by the reviewer 1.

Sincerely,

Reviewer 1

Reviewer’s Comment: The language needs to be improved. It is not clear in the abstract what is the contribution compared to the background they have mentioned mainly because it is not clear what you mean by whole-brain variability.

Authors’ Response:To further clarify what we meant by “whole-brain variability,”werewrote the following part of the Abstract:

In this article, we take a different stance on the analysis of the differential effect of the negative, neutral, and positive affect on the brain functioning in which we look into the whole-brain variability: that is the change in the brain information processing measured in multiple distributed regions. For this purpose, we compute the entropy of individuals’ muti-channel EEG recordings who watched movie clips with differing affect.”

In the current version of the manuscript, the full Abstract reads as follows.

Individuals’ ability to express their subjective experiences in terms of such attributes as pleasant/unpleasant or positive/negative feelings forms a fundamental property of their affect and emotion. However, neuroscientific findings on the underlying neural substrates of the affect appear to be inconclusive with some reporting the presence of distinct and independent brain systems and others identifying flexible and distributed brain regions. A common theme among these studies is the focus on the change in brain activation. As a result, they do not take into account the findings that indicate the brain activation and its information content does not necessarily modulate and that the stimuli with equivalent sensory and behavioural processing demands may not necessarily result in differential brain activation. In this article, we take a different stance on the analysis of the differential effect of the negative, neutral, and positive affect on the brain functioning in which we look into the whole-brain variability: that is the change in the brain information processing measured in multiple distributed regions. For this purpose, we compute the entropy of individuals’ muti-channel EEG recordings who watched movie clips with differing affect. Our results suggest that the whole-brain variability significantly differentiates between the negative, neutral, and positive affect. They also indicate that although some brain regions contribute more to such differences, it is the whole-brain variational pattern that results in their significantly above chance level prediction. These results imply that although the underlying brain substrates for negative, neutral, and positive affect exhibit quantitatively differing degrees of variability, their differences are rather subtly encoded in the whole-brain variational patterns that are distributed across its entire activity.

Reviewer’s Comment: It is better to start with methodology and shows the stand of your work before presenting your result. The only part that is needed to be mentioned before method section is what is used as motivation.

Authors’ Response: We agree with the reviewer on the formatting of the manuscript by first presenting the methodology and then proceeding to the results and discussion. However, we want to bring to the reviewer’s kind attention that we used the manuscript template that wasprovided by the Journal Entropy while preparing our manuscript. As a result, our Results and Discussion Sections proceeded the Materials and Methods Section. We apologize for any inconvenience that the formatting of the manuscript may have caused the reviewerand would like to express our willingness to reformat the ordering of our manuscript, would the Journal’s policy allow for such changes.

Reviewer’s Comment:Adding some results on related methods would be helpful.

Authors’ Response: In the case of DE, we modified the Section5.3. DE Computations, lines358-373,as follows.

We computed DE using the full-length EEG data (i.e., 37000 data points) per channel, per participant, and per affect, using the following procedure. For a given affect (e.g., negative) we accessed its associated five files (i.e., one file for each of the movie clips) of 62-channel EEG recordings one-by-one. For each of these files, we then used each of the EEG channels one-by-one and computed its DE(i.e., one DE for each of the EEG channels). This resulted in five separate sets (i.e., one set for each of the movie clips) of sixty-two DEs (i.e., one DE value for each of the EEG channels). Next, we utilized these five separate sets of 62 DEs and computed the average DE for each channel. For example, for channel F7 (Figure 7 (B)), we had five DEs (i.e., one DE for each movie clips:DEF7movie clip1 , DEF7movie clip2 , DEF7movie clip3 , DEF7movie clip4 , DEF7movie clip5 ).We averaged these five DEs i.e., mean([DEF7movie clip1 , DEF7movie clip2 , DEF7movie clip3 , DEF7movie clip4 , DEF7movie clip5]).In this respect, it is apparent that the ordering of these values have no effect on the computed average DE for F7 in a given affect state (e.g., negative affect) [68]. We repeated this procedure for each participant (i.e., fourteen) and each affect (i.e., negative, neutral, and positive), thereby computing the averaged participants’ brain variability in response to a given affect. This process resulted in 1 × 62 vectors, per affect, per participant, where 62 refers to the number of EEG channels (Figure 7 (B)).

In addition, tofurtherverify the correspondence between the computedDEs associated with negative, neutral, and positive affect states.Specifically,we applied paired (i.e., between every two pairs of affect) Spearman correlation on DEs associated with these affect states.For this purpose, First, we computed the Spearman correlations between the whole-brain DEs of every pair of affect states (i.e., positive versus neutral, positive versus negative, and negative versus neutral). We followed this by computing their bootstrap (10,000 simulation runs) at 95% confidence intervals (i.e., p < .05 significance level). For the bootstrap test, we considered the null hypothesis H0: there was no correlation between every pair of affect states’ whole-brain DEs and tested it against the alternative hypothesis H1: The whole-brain DEs of every pair of affect states correlated significantly. We reported the mean, standard deviation, and the 95.0% confidence interval for these tests. We also computed the p-value of these tests as the fraction of the distribution that was more extreme than the actually observed correlation values. For this purpose, we performed a two-tailed test in which we used the absolute values so that both the positive and the negative correlations were accounted for.

We added this information to Section5.4.1. DE Analyses, lines 386-395. We reported the results associated with this test in Section 2. Results, Subsection 2.1. DE, lines 82-87and also Figure 2 and Table 1,page 4,in the current version of the manuscript.

We also added a new Section4. Limitations and Future Direction where we used these new results for further interpretation of the linear model’s linear accuracy on positive affect as follows (lines 227-273).

Taken together, our findings suggested the significant role of whole-brain variability in response to differential effect of the negative, neutral, and positive affect. Specifically, they indicated that although the underlying brain substrates for negative, neutral, and positive affect exhibited quantitatively differing degrees of variability, their differences were rather subtly encoded in the whole-brain variational patterns that were distributed across its entire activity. Further evidence for the distributed signature of the negative, neutral, and positive affect in the whole-brain variability came from the significant correlation between the DEs of these affect. Interestingly, we also observed that a simple linear classifier was able to distinguish between these affect states with a significantly above chance level accuracy. Furthermore, such an ability was significantly more accurate when the linear model was trained based on the whole-brain variability. In this regard, a puzzling observation was the higher percentage of misclassification between the positive and the negative affect by the model which could not readily be attributed to the sample size since we used a balanced dataset in our analyses (i.e., equal number of samples for each of the negative, neutral, and positive affect, per participant).

Another possible reason behind the observed effect could have been the difference in the shared information between these affect. However, the results of the correlation analyses (i.e., linear measure of mutual information [58]) did not provide any further insight on this matter. For instance, these results could not explain the higher misclassification of the positive than the negative affect as neutral affect by the model, despite the fact that the former shared substantially more information (i.e., substantially higher correlation) with the neutral affect than the latter did. Conversely, the lower shared information also was not sufficient to account for the observed misclassification of the positive affect since the high rate of misclassification between positive and negative should have in principle been the lowest, given their lowest correlation among the three pairwise comparisons. Additionally, we observed that the linear model misclassified the neutral affect as both positive and negative affect in an equal rate although it showed substantially different correlations with these two affect states. Although we also observed that some of the brain regions showed significant differences that was in accord with the whole-brain differential signature of the negative, neutral, and positive affect, the mere use of these regions did not improve the higher percentage of the misclassification between the positive and the negative affect and only resulted in worsening of the model’s performance.

These observations implied that although the negative, neutral, and positive affect appear to share a common distributed neural correlates whose signature can be identified across the whole-brain variability, such an underlying brain dynamical network might be shared differentially among them and that the negative and the positive affect might have more in common (i.e., with respect to the change in the variation in the brain information processing) than with the neutral affect. They also indicated that governing dynamics of the brain responses to these affect might not primarily be explained by linear modeling of such a common distributed neural dynamics. Therefore, future research that takes into account the non-linear dynamics of neural activity in response to negative, neutral, and positive affect is necessary to further extend our understanding of the potential causes of the observed dis/similarities between these affect states.

It is also crucial to note that in the present study we utilized the entropy (i.e., DE) of EEG time series of human subjects for analysis of the whole-brain variability in response to negative, neutral, and positive affect states. In this regard, research identifies a functional interactivity between the brain regions [56] in which signals from individual cortical neurons are shared across multiple areas and thus concurrently contribute to multiple functional pathways [89]. However, our analyses did not take into consideration the potential effect of such flow of information among different brain regions in response to differential affect states while computing the EEG channels’ DEs. Therefore, future research to study the possibility of such potential information flows [90,91] is necessary to better realize the utility of the brain variability in emergence of differential affect states in response to natural stimuli.

We also added a new reference (reference [64] in the current version of the manuscript) for the readers who might be more interested in their interpretation and added the following sentence (lines 355-356, in the current version of the manuscript).

Further information and insights about the mathematical background and interpretation of ADF and KPSS can be found in [6264].”

Reviewer 2 Report

Summary:

The authors investigate whole brain activity (measured with electroencephalography) variability in entropy and how it related to different affective stimuli. They found that these measurements could help differentiate between positive, negative, and neutral affect. They also observe that whole brain entropy variation can predict above chance. Their results suggest whole brain (at least that is measured) distributed networks are important in understanding information encoding.

Notes

My expertise is not directly related to the topics presented within the manuscript; however, I do have experience in related areas and in machine learning. All comments below may not be relevant, and in that case, please ignore and address what is necessary.

Overall the manuscript is exceptionally well written and very easy to follow. I have noted a few small things to be looked into below that the authors can address. Although not an expert in EEG, the background information was well presented and provided enough context to understand the contents. For the most part, the conclusions are humbly put. The statistical analysis was clean and clear. The use of such a simple classifier (logistic regression) is also noteworthy.

I suspect there was an error in the caption of Figure 5 in identifying the subfigures. I think subfigure B and C were mixed up. Assuming this is an error, it is an easy fix.

I personally like seeing p-values reported on as opposed to signifying <0.05, however I understand that this is typical within certain fields.

I would have liked to see a discussion on why the authors think it was easier to correctly classify negative compared to positive affect. This is perhaps a consequence of the sample size (having worked with neuroimaging data, I sympathize and acknowledge that small sample sizes and 1-fold is typical). Regardless, some discussion on this may be valuable.

It is unclear why the authors did not use all three sets of experiments that were included on the dataset. They did explain that some data did not conform to data standards, and as such they eliminated some data, but I feel that the inclusion of more data would have helped with the analysis. Presumably the pipeline would have been identical for all experiment sets. This feels like an easy way to improve results and it is strange that the data was excluded.

Is it typical within the field to ignore the subject affect reporting? Would the self-reported affect not be more representative of how the individual feels? Would the self-reported emotion not be a better proxy for the neural activity? Again, if this is typical within the field, then ignore the comment, however if it is not, this seems strange. Perhaps this should be addressed?

I am not sure the authors can conclude that the whole brain networks are necessary/best based on the data presented. The data presented seems to only suggest that they were more capable when compared to the small subsets of data that they selected. Is there a larger network that is still not the whole brain that would be most effective? I think this can be addressed if the conclusions are rephrased slightly.

Small Notes:
- Abstract "..its information content do not necessarily..." Should the "do" be "does"?

- Intro "... emphasizes on the presence of two..." Should you cut the "on"?
- Intro "...argues that the positive and negative affect are the brain states that are supported by flexible than consistently specific set of brain regions [19]." This reads funny.
- Intro "More specifically, these results do not take into account the findings that indicate the brain activation and its information content do not necessarily modulate [31] and that the stimuli with equivalent sensory and behavioural processing demands may not necessarily result in differential brain activation [32]." Perhaps reword. Also, should it be information content *does*...?
- Intro "...positive and negative affect but also contrast such a variability...", perhaps add a comma after affect.

- Results "... also verified that the DEs associated with the negative state was significantly higher ..." should the was be were?

- Discussion "... the positive affect also hint at the association..." perhaps *hints*
- Discussion "We also observed that the signatures of the negative, neutral, and positive affect were present in the whole-brain variability than designated regions for different affect states." Reads funny.

Author Response

First and foremost, the authors would like to take this opportunity to thank the reviewers and the associate editor for their time and kind consideration to review our manuscript. The comments by the reviewers enabled us to improve the quality of our results and their presentation substantially.

In what follows, we provide point-by-point responses to the comments and concerns raised by the reviewer 2.

Sincerely,

Reviewer 2

Reviewer’s Comment: I suspect there was an error in the caption of Figure 5 in identifying the subfigures. I think subfigure B and C were mixed up. Assuming this is an error, it is an easy fix.

Authors’ Response: We thank the reviewer for spotting this typo. We applied a number of changes to this figure (Figure 6, page 7,in the current version of the manuscript). They are as follow.

We color-coded the confusion matrices and adjusted the font size of panel E (Figure 6, panel C, page 7,in the current version of the manuscript).We also realized that the use of subplots B and D were redundant and might confuse the reader. Therefore, we removed these subplots from this figure andmodifieditscaption as follows.

Figure 6. Linear model’s confusion matrices associated with its prediction accuracy in negative, neutral, and positive affect in 1-holdout setting using (A) whole-brain DEs and (B) subset of channels with the similar pattern of significant differences as whole-brain DEs. These results show the model’s performance after data from every participant was used for testing. Correct predictions, per affect, are the diagonal entries (blue) of these tables and the off-diagonal entries (red) show the percentage of each of affect that was misclassified (e.g., positive affect misclassified as negative affect). (C) Two-sample bootstrap test of significance (10,000 simulation runs) at 99.0% (i.e., p < .01) confidence interval between the accuracy of linear model using the whole-brain DEs versus subset of channels with the similar pattern of significant differences as whole-brain DEs.”

To provide further insight aboutthe linear model’s performance, we addeda new table (Table 3, page 8,in the current version of the manuscript) that summarizedthe precision, recall, and F1-score of the model.

To reflect these changes in the text, we modified the manuscript’s content (lines123-151in the current version of the manuscript) as follows:

Figure6(A) shows the linear model’s prediction accuracy in negative, neutral, and positive affect in 1-holdout setting using whole-brain DEs. In this setting, we considered a single participant’s positive, negative, and neutral affect data as test set and used the remaining participants’ data for training the linear model. We then tested the model’s performance on the holdout test data. We repeated this procedure for every participant. Subplot (A) in Figure 6 indicates that the use of whole-brain DEs to quantify the brain variability of the participants enabled the linear model to predict their negative, neutral, and positive affect with significantly above chance level (chance level accuracy ≈ 33.33%, given the three-affect settings). One-sample bootstrap test of significance (10,000 simulation runs) at 95.0% (i.e., p < .05) confidence interval verified this observation (M = 78.52, SD = 6.45, CI95% = [61.90 88.10]). We also observed that these predictions were substantially higher in the case of negative and neutral than the positive affect. An inspection of Figure 6 (A) reveals that a relatively larger number of participants’ positive affect was misclassified as negative affect by the linear model. Table 3, top row, summarizes the precision, recall, and F1-score associated with the linear model’s performance in negative, neutral, and positive affect in 1-holdout setting using whole-brain DEs.

We further checked the linear model’s performance using the subset of channels with the similar pattern of significant differences as whole-brain DEs (Figure 6(B)). Our analyses (for details, see Appendix A) identified that these channels were in the frontal (F3, FZ, F4), frontocentral (FCZ), central (C5, C3, C1, C2, C4), centroparietal (CP3, CP1, CP2), parietal (P3, POZ), and occipital (CB1, OZ) regions. Figure 6 (B) corresponds to the 1-holdout setting using this subset of participants’ channels. This subplot reveals a substantial reduction of linear model’s accuracy in comparison with the case in which we used the whole-brain DEs. Wilcoxon test identified a significant difference between the linear model’s accuracy in these two settings (p <.001, W(998) = 7.23, r = .23) which we furtherverified using two-sample bootstrap test of significance (10,000 simulation runs) at 99.0% (i.e., p < .01) confidence interval (Figure 6 (C)) (Mdi f f erence = 12.54, SDdi f f erence = 1.63, CIdi f f erence = [8.27 16.67]) (see Appendix C for randomized 500 simulation runs of these results). Table 3, bottom row, summarizes the precision, recall, and F1-score associated with the linear model’s performance in negative, neutral, and positive affect in 1-holdout setting using this subset of channels with the similar pattern of significant differences as whole-brain DEs.”

We applied a similar modification to Figure A4. (page23,in the current version of the manuscript). It reads as follows.

Figure A4. Confusion matrices associated with randomized affect prediction based on individual’s brain variability by the linear model in 1-holdout setting (500 simulation runs, per affect) using (A) whole-brain and (B) subset of channels with the similar pattern of significant differences as whole-brain DEs. These results show the model’s performance after data from every participant was used for testing. Model’s accuracy for correct and incorrect predictions are the diagonal and off-diagonal entires.

We also added a new table (Table A4., page 23,in the current version of the manuscript) that summarizedthe linear model’s precision, recall, and F1-score associated with this randomized setting. To reflect these changes, we modified its content (Appendix C Linear Model’s Prediction - Randomized Case, Lines 414-434) as follows:

FigureA4shows the linear model’s prediction of negative, neutral, and positive affect in a randomized setting (for both training scenarios) in which we first randomly selected an individual and then picked, at random, only one of this selected individual’s negative, neutral, or positive affect data for 1-holdout testing. We then used the remaining data that did not include any of the randomly selected individual’s affect data for training. To guarantee an unbiased estimate of the linear model’s prediction, we continued this random selection until every affect was selected 500 times. This differed from the first 1-holdout setting (reported in the main manuscript) in which we used data pertinent to all three affect of the selected individual for 1-holdout test.

The confusion matrix associated with these 500 rounds of individual’s random selection for 1-holdout test is shown in Figure A4(A). Similar to the case of single-trial, per participant (Section 6.2 in the main manuscript), the use of whole-brain variability enabled the linear model to predict the negative, neutral, and positive affect with the accuracy that was significantly above chance level (chance level accuracy ≈ 33.33%, given the three-affect setting). In addition, we also observed that these predictions were substantially higher in the case of negative and neutral than the positive affect. An inspection of Figure A4(B) reveals that a relatively large number of participants’ positive affect was (similar to the case of single-trial, per participant; Section 6.2 in the main manuscript) misclassified as representing their negative affect (i.e., 141 out of 183 misclassified positive state in Figure A4 (B) bottom row entry).

TableA4summarizes the precision, recall, and F1-score of the linear model’s performance associated with this randomized setting. A comparison between entries of this table and Table 3reveals that these model’s metrics remained stable.”

Reviewer’s Comment: I personally like seeing p-values reported on as opposed to signifying <0.05, however I understand that this is typical within certain fields.

Authors’ Response: We also agree with the reviewer on this matter. We followed the format of reporting the p-value (to the best of our knowledge) that is generally practiced bythe Journal Entropy. However, we are able to change them in case the reviewer considers it to be necessary andthe Journal’s policy allows for such changes.

Reviewer’s Comment:I would have liked to see a discussion on why the authors think it was easier to correctly classify negative compared to positive affect. This is perhaps a consequence of the sample size (having worked with neuroimaging data, I sympathize and acknowledge that small sample sizes and 1-fold is typical). Regardless, some discussion on this may be valuable.

Authors’ Response:This is indeed an important observation that requires further investigation.Althoughwe agree with the reviewer’s viewon the potential effect ofthesmall sample size, we also think that it cannot be considered as the mainreason for the observed difference in the model’s accuracy for negative and positive affect.Our speculation is due to the factthat the data associated with three affect were balanced i.e., every participant had data for each of the negative, neutral, and positive affect. Therefore, the sample size should have affected the overall performance of the model and notonly the positive affect.

One possible reason behind the observed effect could havebeenthe difference in the shared information between these affect. To further investigate this possibility we added a newcorrelation (i.e., linear mutual information [58])analysis to this version of the manuscript. Specifically, wecomputed the Spearman correlations between the whole-brain DEs of every pair of affect states (i.e., positive versus neutral, positive versus negative, and negative versus neutral). We followed this by computing their bootstrap (10,000 simulation runs) at 95% confidence intervals (i.e., p < .05 significance level). For the bootstrap test, we considered the null hypothesis H0: there was no correlation between every pair of affect states’ whole-brain DEs and tested it against the alternative hypothesisH1: The whole-brain DEs of every pair of affect states correlated significantly. We reported the mean, standard deviation, and the 95.0% confidence interval for these tests. We also computed the p-value of these tests as the fraction of the distribution that was more extreme than the actually observed correlation values. For this purpose, we performed a two-tailed test in which we used the absolute values so that both the positive and the negative correlations were accounted for. We added this information to Section5.4.1. DE Analyses, lines 386-395. We reported the results associated with this test in Section 2. Results, Subsection 2.1. DE, lines 82-87and also Figure 2 and Table 1,page 4,in the current version of the manuscript.

However, these results also did not provide any further insight on this matter. For instance, theycould not explain the higher misclassification of the positive than the negative affect as neutral affect by the model, despite the fact that the former shared substantially more information (i.e., substantially higher correlation) with the neutral affect than the latter. Conversely, the lower shared information also was not sufficient to account for the observed misclassification of the positive affect since the high rate of misclassification between positive and negative should have in principle been the lowest, given their lowest correlation among the three pairwise comparison. Furthermore, the model misclassified the neutral affect as both positive and negative affect in an equal rateeventhoughwe observedsubstantially different correlations with these two affect states. Although we also observed that some of the brain regions showed significant differences that was in accord with the whole-brain differential signature of the negative, neutral, and positive affect, the mere use of these regions did not improve the higher percentage of the misclassification between the positive and the negative affect and only resulted in worsening of the model’s performance.

These observations impliedthat although the negative, neutral, and positive affect appear to share a common distributed neural correlateswhose signature can be identified across the whole-brain variability, such an underlying brain dynamical network might be shared differentially among them and that the negative and the positive affect might have more in common (i.e., with respect to the change in the variation in the brain information processing) than with the neutral affect. They also indicatedthat governingdynamics ofthe brain responses to these affect might not primarily be explained by linearmodelingof such a commondistributed neural dynamics.Therefore, future research that takes into account the non-linear dynamics of neural activity in response to negative, neutral, and positive affect is necessary to further our understanding of the potential causes of the observed dis/similarities between these affect states.

In the current version of the manuscript, we added a new SectionLimitations and Future Direction (pages 9-11)in which we included the followingdiscussion (lines227-273).

Taken together, our findings suggested the significant role of whole-brain variability in response to differential effect of the negative, neutral, and positive affect. Specifically, they indicated that although the underlying brain substrates for negative, neutral, and positive affect exhibited quantitatively differing degrees of variability, their differences were rather subtly encoded in the whole-brain variational patterns that were distributed across its entire activity. Further evidence for the distributed signature of the negative, neutral, and positive affect in the whole-brain variability came from the significant correlation between the DEs of these affect. Interestingly, we also observed that a simple linear classifier was able to distinguish between these affect states with a significantly above chance level accuracy. Furthermore, such an ability was significantly more accurate when the linear model was trained based on the whole-brain variability. In this regard, a puzzling observation was the higher percentage of misclassification between the positive and the negative affect by the model which could not readily be attributed to the sample size since we used a balanced dataset in our analyses (i.e., equal number of samples for each of the negative, neutral, and positive affect, per participant).

Another possible reason behind the observed effect could have been the difference in the shared information between these affect. However, the results of the correlation analyses (i.e., linear measure of mutual information [58]) did not provide any further insight on this matter. For instance, these results could not explain the higher misclassification of the positive than the negative affect as neutral affect by the model, despite the fact that the former shared substantially more information (i.e., substantially higher correlation) with the neutral affect than the latter did. Conversely, the lower shared information also was not sufficient to account for the observed misclassification of the positive affect since the high rate of misclassification between positive and negative should have in principle been the lowest, given their lowest correlation among the three pairwise comparisons. Additionally, we observed that the linear model misclassified the neutral affect as both positive and negative affect in an equal rate although it showed substantially different correlations with these two affect states. Although we also observed that some of the brain regions showed significant differences that was in accord with the whole-brain differential signature of the negative, neutral, and positive affect, the mere use of these regions did not improve the higher percentage of the misclassification between the positive and the negative affect and only resulted in worsening of the model’s performance.

These observations implied that although the negative, neutral, and positive affect appear to share a common distributed neural correlates whose signature can be identified across the whole-brain variability, such an underlying brain dynamical network might be shared differentially among them and that the negative and the positive affect might have more in common (i.e., with respect to the change in the variation in the brain information processing) than with the neutral affect. They also indicated that governing dynamics of the brain responses to these affect might not primarily be explained by linear modeling of such a common distributed neural dynamics. Therefore, future research that takes into account the non-linear dynamics of neural activity in response to negative, neutral, and positive affect is necessary to further extend our understanding of the potential causes of the observed dis/similarities between these affect states.

It is also crucial to note that in the present study we utilized the entropy (i.e., DE) of EEG time series of human subjects for analysis of the whole-brain variability in response to negative, neutral, and positive affect states. In this regard, research identifies a functional interactivity between the brain regions [56] in which signals from individual cortical neurons are shared across multiple areas and thus concurrently contribute to multiple functional pathways [89]. However, our analyses did not take into consideration the potential effect of such flow of information among different brain regions in response to differential affect states while computing the EEG channels’ DEs. Therefore, future research to study the possibility of such potential information flows [90,91] is necessary to better realize the utility of the brain variability in emergence of differential affect states in response to natural stimuli.

Reviewer’s Comment:Reviewer’s Comment: It is unclear why the authors did not use all three sets of experiments that were included on the dataset. They did explain that some data did not conform to data standards, and as such they eliminated some data, but I feel that the inclusion of more data would have helped with the analysis. Presumably the pipeline would have been identical for all experiment sets. This feels like an easy way to improve results and it is strange that the data was excluded.

Authors’ Response: The main goal of the present study was to verify whether the underlying brain substrates for negative, neutral, and positive affect were rather subtly encoded in the whole-brain variational patterns that were distributed across its entire activity. As a result, we primarily focused on the statistical analyses ofthe whole-brain variability in terms of its distributed information processing (i.e., its entropy)and used the linear model as a supportive evidence that showed that whole-brain variability in fact resulted in better representation of these three affect states than the use ofselected brain regions that also showed similar statistically significant differences.

In this respect, including several measurements of the same subjects that watched the same movie clips in the same order in multiple days would have only complicated the interpretation of the results due to suchissuesas multiple-comparison as well aspotentiallyconfounding factors suchasredundant information (e.g., assuming no substantial change in individuals’ brain functioning, it is plausible to expectthat their brain not to respond significantly differentlyto the same stimuli that was presented to them in different days).Therefore, we decided toreducethe possibility of occurrence of such issues and confounders by only including one out of three sessions of each of the participants.

Considering the case of the linear model, we were also concerned with the potential effect of multiple measurements of same individuals on the credibility of the observed accuracy of the trained model. This concern was the main reason that we decided to exclude all three negative, neutral, and positive affect measurement of every individual who was considered as the test subject for the linear model. More specifically, by doing so we made all effort to ensure that all potential resemblances between train and test data for the model were excluded. In this respect, adding other sessions of the individuals could not have changed the overall accuracy of the modelsince we would have needed to separate them all (i.e., per test individual) from the training data.

However,we also agree with the reviewer’s comment on the benefit of larger sample size. In fact, this becomes a crucial factor in the scenarios in which the goal is to train personalized model for the individuals (e.g., personalized socially-assistive robots [92]).Therefore, we added the following discussion to the newly added Section4. Limitations and Future Direction(lines274-294,in the current version of the manuscript).

The main goal of the present study was to verify whether the underlying brain substrates for negative, neutral, and positive affect were rather subtly encoded in the whole-brain variational patterns that were distributed across its entire activity. As a result, we primarily focused on the statistical analyses of the whole-brain variability in terms of its distributed information processing (i.e., its entropy) and used the linear model as a supportive evidence that showed that whole-brain variability in fact resulted in better representation of these three affect states than the use of selected brain regions that also showed similar statistically significant differences. In this respect, including several measurements of the same subjects that watched the same movie clips in the same order in multiple days would have only complicated the interpretation of the results due to such issues as multiple-comparison as well as potentially confounding factors such as redundant information (e.g., assuming no substantial change in individuals’ brain functioning, it is plausible to expect thattheir brain not to respond significantly differently to the same stimuli that was presented to them in different days). Therefore, we decided to reduce the possibility of occurrence of such issues and confounders by only including one out of three sessions of each of the participants. However, such multiple measurements can benefit the future research by providing an opportunity to test for the reproducibility of the current results. For instance, one can compute their respective whole-brain variability using their respective DEs and compare their corresponding neural substrates with the results in the present study. They can also be utilized as one larger test set (i.e., all together) to verify whether the linear model in this study can preserve its accuracy on this new data. The latter scenario can become a valuable testbed for the cases in which training personalized models for the individuals is desirable (e.g., personalized socially-assistive robots [92]).”

In addition, we also emphasized the need for future study with larger sample sizes for more informed conclusion on our findings as follows (Section 4. Limitations and Future Direction, lines 295-299).

Last, although neuropsychological findings indicate that the individuals’ ability to experience pleasant/unpleasant feelings to express these subjective mental states in terms of such attributes as positive or negative [25] to be the unifying and common concept across cultures [11,12], future research that includes a larger human sample as well as different age groups along with more cultural diversity is necessary for drawing a more informed conclusion on the findings that were presented in this article.

Reviewer’s Comment: Is it typical within the field to ignore the subject affect reporting? Would the self-reported affect not be more representative of how the individual feels? Would the self-reported emotion not be a better proxy for the neural activity? Again, if this is typical within the field, then ignore the comment, however if it is not, this seems strange. Perhaps this should be addressed?

Authors’ Response: We believe this comment is due to our manuscript failing to provide proper and sufficient explanation about the procedure by which the affect scores and self-assessment responses were collected and used in this dataset. We apologize for this shortcoming.

In what follows, we addressed the concern of the reviewer in two steps: 1) How elicited affect of the movie clips were verified 2) What were the content of the self-assessment questions and how they were used.

How elicited affect of the movie clips were verified: To ensure that the selected movie clips indeed elicited the targeted affect, a preliminary study [60] was conducted where twenty participants were asked to assess a pool of movie clipsin afive-point scale. Based on this study, fifteen movie clips (i.e., five clips for eachnegative, neutral, and positive affect) whose average score were >= 3 and ranked in the top 5 in each affect category, were chosen.

We have addedthis information tothe current version of themanuscript,Section5.1. The Dataset, lines 323-327.

What were the content of the self-assessment questions and how they were used: At the end of each movie clip, the participants were asked to answer three questions that followed the Philippot [61]. These questions were the type of emotion that the participants actually felt while watching the movie clips, whether they watched the original movies from which the clips were taken, and whether they understood the content of those clips. The participants responded to these three questions by scoring them in the scale of 1 to 5. Subsequently, the trials whose scores were below 3 were discarded since they failed to elicit the targeted affect. In other words, the self-assessment were not reported as a part of experiment’s materials but primarily used to check the status of the recorded brain activity in response to targeted affect.

We have addedthis information tothe current version of themanuscript,Section5.1. The Dataset, lines 307-313and also the caption of Figure 7 (page 12 in the current version of the manuscript).

In essence, the authors of SEED [59,60] did not use the participants’ self-assessed responses to label/categorize the movie clips. On the contrary, they first ensured that their selected movie clips in fact elicited the targeted affect and then used the self-assessed responses to discard the trials that might potentially not best represent the type of sensation/affect their movie clipsaimed for. In this regard, we also want to bring to the reviewer’s kindconsiderationthe new findings that identified that self-assessed responses are in fact not the best indicator of induced effect onthe brain activity. Precisely,MacDuffieet al. [1] used asample of 1256 human subjects and showed that the participants’ self-ratings “were uncorrelated with actual … activity measured via BOLD fMRI.On the other hand, they identified these ratings to be the bestpredictorsof “subjective distress across a variety of self-report measures.They concluded that their findings suggest that such ratings “while unrelated to measured neural activation may be informative indicators of psychological functioning.

[1]MacDuffie, K.E., Knodt, A.R., Radtke, S.R., Strauman, T.J. and Hariri, A.R., 2019. Self-rated amygdala activity: an auto-biological index of affective distress. Personality Neuroscience,volume2,2019.

Reviewer’s Comment:I am not sure the authors can conclude that the whole brain networks are necessary/best based on the data presented. The data presented seems to only suggest that they were more capable when compared to the small subsets of data that they selected. Is there a larger network that is still not the whole brain that would be most effective? I think this can be addressed if the conclusions are rephrased slightly.

Authors’ Response:Although we are not sure, we think that the reviewer referred to the Discussion Section, lines 170-173. Therefore, we modified this content (3. Discussion,lines 208-213,in the current version of the manuscript)as follows.

Our results suggested that the whole brain networks were more capable when compared to the brain regions that corresponded to these subset of channels. These results appeared to extend the fMRI-based findings by Saarimäki [80] that indicated that the anatomically distributed variational patterns of brain activity contained the most accurate neural signature of individuals’ mental states that underly their discrete emotions.

We would also like to ask the reviewer to please let us know if there is any other part of the manuscript that may require similar modification.

Reviewer’s Comment:- Abstract "..its information content do not necessarily..." Should the "do" be "does"?

Authors’ Response: We corrected this typo. Furthermore, to make the content of Abstract more clear (e.g., what did we mean by “whole-brain variability”) we modified the Abstract as follows.

Individuals’ ability to express their subjective experiences in terms of such attributes as pleasant/unpleasant or positive/negative feelings forms a fundamental property of their affect and emotion. However, neuroscientific findings on the underlying neural substrates of the affect appear to be inconclusive with some reporting the presence of distinct and independent brain systems and others identifying flexible and distributed brain regions. A common theme among these studies is the focus on the change in brain activation. As a result, they do not take into account the findings that indicate the brain activation and its information content does not necessarily modulate and that the stimuli with equivalent sensory and behavioural processing demands may not necessarily result in differential brain activation. In this article, we take a different stance on the analysis of the differential effect of the negative, neutral, and positive affect on the brain functioning in which we look into the whole-brain variability: that is the change in the brain information processing measured in multiple distributed regions. For this purpose, we compute the entropy of individuals’ muti-channel EEG recordings who watched movie clips with differing affect. Our results suggest that the whole-brain variability significantly differentiates between the negative, neutral, and positive affect. They also indicate that although some brain regions contribute more to such differences, it is the whole-brain variational pattern that results in their significantly above chance level prediction. These results imply that although the underlying brain substrates for negative, neutral, and positive affect exhibit quantitatively differing degrees of variability, their differences are rather subtly encoded in the whole-brain variational patterns that are distributed across its entire activity.

Reviewer’s Comment:- Intro "... emphasizes on the presence of two..." Should you cut the "on"?

Authors’ Response: We deleted the “on”

Reviewer’s Comment:- Intro "...argues that the positive and negative affect are the brain states that are supported by flexible than consistently specific set of brain regions [19]." This reads funny.

Authors’ Response: We changed this sentence to read as (lines 35-37,in the current version of the manuscript):

On the other hand, the affective workspace hypothesis [8] argues that the positive and negative affect are the brain states that are supported by flexible and distributed brain regions [19].

Reviewer’s Comment:- Intro "More specifically, these results do not take into account the findings that indicate the brain activation and its information content do not necessarily modulate [31] and that the stimuli with equivalent sensory and behavioural processing demands may not necessarily result in differential brain activation [32]." Perhaps reword. Also, should it be information content *does*...?

Authors’ Response: We changed “do” to “does”.

Reviewer’s Comment:- Intro "...positive and negative affect but also contrast such a variability...", perhaps add a comma after affect.

Authors’ Response: We added a comma after “affect”

Reviewer’s Comment:- Results "... also verified that the DEs associated with the negative state was significantly higher ..." should the was be were?

Authors’ Response: We corrected this mistake.

Reviewer’s Comment:- Discussion "... the positive affect also hint at the association..." perhaps *hints*

Authors’ Response: We changed “hint” to “hinted” given the past tense of this paragraph.

Reviewer’s Comment:- Discussion "We also observed that the signatures of the negative, neutral, and positive affect were present in the whole-brain variability than designated regions for different affect states." Reads funny.

Authors’ Response: We changed this sentence as follow (lines 175-176,in the current version of the manuscript).

We also observed that the signatures of the negative, neutral, and positive affect were present in the whole-brain variability.

Reviewer 3 Report

The authors of this study try to show that with whole-brain variability reflected by its entropy it is possible to significantly distinguish when participants from a previous study were observing clips meant to elicit different affective states: positive, neutral and negative. However, we don’t know how effective these clips are to elicit affective responses as these data were not analyzed / reported. More importantly, there are possible statistical issues with averaging the entropy values over the affective clips first, and then over channels, which could render the main results incorrect. This is the main concern in this review so authors should reply thoroughly to comment #24.

Comments:

#1 Line 11: “Our results suggest that the whole-brain variability significantly differentiates between the negative, neutral, and positive affect”. This is no accurate. The results, if statistically sound, suggest that whole-brain variability differentiates viewing different movies. The self- reported states need to be used to support this claim, wither by showing a linear relation with entropy or by showing that there’s a large and significant effect between movie and reported affect (in the correct direction i.e., movie for positive valence is significantly related with positive self-reported affect).

#2: Some sentences are not clear. Example:

line 44 page 2: More specifically, these results do not take into account the findings that indicate the (should be “that”) brain activation and its information content do not necessarily modulate

#3: Lines 66 to 70. No clear why these results go in favor of the affective workspace hypothesis and not the bivalent.

#4: Line 73: How many participants? Are these maps averages of all participants?

#5: Line 76: figure 1 Letters B and C are not correct. Positive is C and neutral B, according to Figure 1 legend.

#6: Line 82: Significant difference between negative, neutral and positive affect on what? Each electrode’s DE or whole brain DE?

#7: Line 82: How was this test performed? Why a non-parametric test?

#8: Line 87: A test of significance of what?

#9: Line 87, Figure 3, Table 1: Why do we need this paired bootstrap test? Shouldn’t the previous post-hoc Wilcoxon test be enough? Is it because it’s paired? If so, it should be briefly explained why this is relevant.

#10: Line 115: Please state size of training and holdout samples.

#11: Line 115: what exactly is a “1-holdout setting”? (training and test set?)

#12: Line 116: Does this mean there was a leave one out cross validation with the training sample? Not clear.

#13: Figure 5: confusion matrices could benefit from a color scale (blue for the diagonal, red for the off-diagonal items, light for low values, stronger for higher values). Why only the panels C and D are called confusion matrices? Aren’t A and B also? Increase font size of panel E…

#14: 118: So this is a leave one out cross validation?

#15: Line 118-119: So, in line 114 you said “Figure 5 (A) shows the linear model’s prediction accuracy in negative, neutral, and positive 115 affect in 1-holdout setting using whole-brain DEs”. This was for one participant only? Then in these lines you say “he confusion matrix associated with the 1-holdout setting in the case of whole-brain DEs is shown in Figure 5 (B)”. The difference is that panel B is for the entire leave one out cross validation? This isn’t totally clear and the readers need to know from which samples these accuracies are computed from, the size of the test samples, etc…

#16: Line 121 to 128: Why a bootstrap and not a p-value obtained from a binomial distribution?

#17: Line 212: No linear drift removal?

#18: Figure 6. Panel A) “seconds” is not well written. Are the subjective self-reported emotions associated with DE? Also, are they associated with the supposed elicited emotions?

#19: Line 224: state that length is data points and not seconds (even though it’s obvious).

#20: Line 226: Two lines before the minimum number of observed data points was 37001 and not 37000…

#21: Line 232: Given the low N, why was this participant promptly excluded? Couldn’t the data pre-processed? Detect where the unit root is; apply a derivative to remove unit root, etc…

#22: Line 235: Bad channel removal and replacement (by neighbors average for example) is common practice in some EEG analysis. Or another procedure to remove unit roots. Why isn’t it applied here? Replacing an entire session with the subsequent (second or third) seems to add more bias than removing the channel. Could you explain why this was done?

#23: Line 241: were self-reported ratings significantly different between (and matching) the different emotional clips? It’s important that the self-reported emotions match the affect states that the clips are supposed to elicit.

#24: Lines 253 to 259: IMPORTANT. This is the most relevant point regarding this paper and if not approached properly, the paper shouldn’t be published, in this reviewer’s opinion. It is clear from the description that the DEs are first averaged by condition (i.e. affective state) which results in a [1x 62 x affect x participant] matrix as stated in lines 258 and 259. The problem is that the order of averaging matters, as has been demonstrated by a response to a result of a recent BCI controversial study (https://doi.org/10.1371/journal.pbio.2004750). Here, the author shows that averaging first by trials and then by channels removes the variance over trials resulting in very low variance between channels which will increase the chance of finding a false significant difference. Furthermore the author of the cited work writes:

This does not mean that averages should not be used for statistical analysis, but that the order of averaging matters. If the data are first averaged over channels and then over trials, the variance over sessions is retained.

And:

Regarding the averaging over channels, it should also be mentioned that averaging across channels is generally not recommended. Although it is not incorrect, averaging channels can reduce or even cancel out an effect in the data if the signal of interest is highly localized or shows different patterns across different brain areas. For this reason, an analysis treating each channel independently (as in S1 Text) is recommended.

Given these points, and that the main results of the paper under review are on the averaged DEs over channels, I have to ask the authors to critically explain if and why their methodology falls within or outside the described problem. I also have to ask the editor to take this comment with high relevance since we shouldn’t allow results stemming from already known incorrect methodology to be published. I should also state that this can be a consequence from this format of relegating the methods to the end, instead of having them before the Result section. The Methods section should be as relevant as the Results section in journals heavy on methodological work.

Also, it’s not stated here how the whole-brain DE is computed So I’m assuming an average over the affect dimension of the [1 x 62 x affect x participant] matrix.

#25: Line 167: replace “comprised of” with “comprises”

#26: Line 267: Why non-parametric?

#27: Line 269: Why the further validation?

Author Response

First and foremost, the authors would like to take this opportunity to thank the reviewers and the associate editor for their time and kind consideration to review our manuscript. The comments by the reviewers enabled us to improve the quality of our results and their presentation substantially.

In what follows, we provide point-by-point responses to the comments and concerns raised by the reviewer 3.

Sincerely,

Reviewer 3

Prior to providing our responses, wewant to bring to the reviewer’s kind attention that, we used the manuscript template that wasprovided by the Journal Entropy while preparing our manuscript. As a result, our Results and Discussion Sections proceeded the Materials and Methods Section. We apologize for any inconvenience that the formatting of the manuscript may have caused the reviewerand would like to express our willingness to reformat the ordering of our manuscript, would the Journal’s policy allow for such changes.

Reviewer’s Comment: #1 Line 11: “Our results suggest that the whole-brain variability significantly differentiates between the negative, neutral, and positive affect”. This is no accurate. The results, if statistically sound, suggest that whole-brain variability differentiates viewing different movies. The self- reported states need to be used to support this claim, wither by showing a linear relation with entropy or by showing that there’s a large and significant effect between movie and reported affect (in the correct direction i.e., movie for positive valence is significantly related with positive self-reported affect).

Authors’ Response:We believe this issue is due to our manuscript failing to provide proper and sufficient explanation about the procedure by which the affect scores and self-assessment responses were collected and used in this dataset. We apologize for this shortcoming.

In what follows, we addressed the concern of the reviewer in three steps: 1) How elicited affect of the movie clips were verified 2) What were the content of the self-assessment questions and how they were used 3) Using Spearman correlation analysis to further validate the correspondence between the negative, neutral, and positive affect states as quantified by their respective entropies (i.e., DEs in our manuscript).

How elicited affect of the movie clips were verified: To ensure that the selected movie clips indeed elicited the targeted affect, a preliminary study [60] was conducted where twenty participants were asked to assess a pool of movie clipsin afive-point scale. Based on this study, fifteen movie clips (i.e., five clips for eachnegative, neutral, and positive affect) whose average score were >= 3 and ranked in the top 5 in each affect category, were chosen.

We have addedthis information tothe current version of themanuscript,Section5.1. The Dataset, lines 323-327.

What were the content of the self-assessment questions and how they were used: At the end of each movie clip, the participants were asked to answer three questions that followed the Philippot [61]. These questions were the type of emotion that the participants actually felt while watching the movie clips, whether they watched the original movies from which the clips were taken, and whether they understood the content of those clips. The participants responded to these three questions by scoring them in the scale of 1 to 5. Subsequently, the trials whose scores were below 3 were discarded since they failed to elicit the targeted affect. In other words, the self-assessment were not reported as a part of experiment’s materials but primarily used to check the status of the recorded brain activity in response to targeted affect.

We have addedthis information tothe current version of themanuscript,Section5.1. The Dataset, lines 307-313and also the caption of Figure 7 (page 12 in the current version of the manuscript).

In essence, the authors of SEED [59,60] did not use the participants’ self-assessed responses to label/categorize the movie clips. On the contrary, they first ensured that their selected movie clips in fact elicited the targeted affect and then used the self-assessed responses to discard the trials that might potentially not best represent the type of sensation/affect their movie clipsaimed for. In this regard, we also want to bring to the reviewer’s kindconsiderationthe new findings that identified that self-assessed responses are in fact not the best indicator of induced effect onthe brain activity. Precisely,MacDuffieet al. (reference[1]below)used asample of 1256 human subjects and showed that the participants’ self-ratings “were uncorrelated with actual … activity measured via BOLD fMRI.On the other hand, they identified these ratings to be the bestpredictorsof “subjective distress across a variety of self-report measures.They concluded that their findings suggest that such ratings “while unrelated to measured neural activation may be informative indicators of psychological functioning.

[1]MacDuffie, K.E., Knodt, A.R., Radtke, S.R., Strauman, T.J. and Hariri, A.R., 2019. Self- rated amygdala activity: an auto-biological index of affective distress. Personality Neuroscience,volume2,2019.

Using Spearman correlation analysis to further validate the correspondence between the negative, neutral, and positive affect states as quantified by their respective entropies (i.e., DEs in our manuscript):Although there was no self-assessment of the participants available, we found the reviewer’s comment instructive and therefore added this new analysisto verify the correspondence between the computed DEs associated with negative, neutral, and positive affect states. Specifically, we applied paired (i.e., between every two pairs of affect) Spearman correlation on DEs associated with these affect states.For this purpose, wefirstcomputed the Spearman correlations between the whole-brain DEs of every pair of affect states (i.e., positive versus neutral, positive versus negative, and negative versus neutral). We followed this by computing their bootstrap (10,000 simulation runs) at 95% confidence intervals (i.e., p < .05 significance level). For the bootstrap test, we considered the null hypothesis H0: there was no correlation between every pair of affect states’ whole-brain DEs and tested it against the alternative hypothesisH1: The whole-brain DEs of every pair of affect states correlated significantly. We reported the mean, standard deviation, and the 95.0% confidence interval for these tests. We also computed the p-value of these tests as the fraction of the distribution that was more extreme than the actually observed correlation values. For this purpose, we performed a two-tailed test in which we used the absolute values so that both the positive and the negative correlations were accounted for.

We added this information to Section 5.4.1. DE Analyses, lines 386-395. We reported the results associated with this test in Section 2. Results, Subsection 2.1. DE, lines 82-87and also Figure 2 and Table 1,page 4,in the current version of the manuscript.

We also added a new Section4. Limitations and Future Direction where we used these new results for further interpretation of the linear model’s accuracy on positive affect as follows (lines 227-299).

Another possible reason behind the observed effect could have been the difference in the shared information between these affect. However, the results of the correlation analyses (i.e., linear measure of mutual information [58]) did not provide any further insight on this matter. For instance, these results could not explain the higher misclassification of the positive than the negative affect as neutral affect by the model, despite the fact that the former shared substantially more information (i.e., substantially higher correlation) with the neutral affect than the latter did. Conversely, the lower shared information also was not sufficient to account for the observed misclassification of the positive affect since the high rate of misclassification between positive and negative should have in principle been the lowest, given their lowest correlation among the three pairwise comparisons. Additionally, we observed that the linear model misclassified the neutral affect as both positive and negative affect in an equal rate although it showed substantially different correlations with these two affect states. Although we also observed that some of the brain regions showed significant differences that was in accord with the whole-brain differential signature of the negative, neutral, and positive affect, the mere use of these regions did not improve the higher percentage of the misclassification between the positive and the negative affect and only resulted in worsening of the model’s performance.

These observations implied that although the negative, neutral, and positive affect appear to share a common distributed neural correlates whose signature can be identified across the whole-brain variability, such an underlying brain dynamical network might be shared differentially among them and that the negative and the positive affect might have more in common (i.e., with respect to the change in the variation in the brain information processing) than with the neutral affect. They also indicated that governing dynamics of the brain responses to these affect might not primarily be explained by linear modeling of such a common distributed neural dynamics. Therefore, future research that takes into account the non-linear dynamics of neural activity in response to negative, neutral, and positive affect is necessary to further extend our understanding of the potential causes of the observed dis/similarities between these affect states.

Reviewer’s Comment: #2: Some sentences are not clear. Example: line 44 page 2: More specifically, these results do not take into account the findings that indicate the (should be “that”) brain activation and it information content do not necessarily modulate

Authors’ Response: We modified this sentence.In addition, we further audited and proofread our manuscript to ensurethat its content was clear and easy to follow.

Reviewer’s Comment: #3: Lines 66 to 70. No clear why these results go in favor of the affective workspace hypothesis and not the bivalent.

Authors’ Response:To address the reviewer’s comment, we first removedthe interpretation of our suggestion from this paragraph and moved it toDiscussion Section. In the current version of the manuscript, this originalparagraph in theIntroduction Section(lines 65-71) reads as follows.

Our results suggest that the whole-brain variability significantly differentiates between the negative, neutral, and positive affect. They also indicate that although some brain regions contribute more to such differences, it is the whole-brain variational pattern that results in significantly above chance level prediction of the negative, neutral, and positive affect. Specifically, they identify that although the underlying brain substrates for negative, neutral, and positive affect exhibit quantitatively differing degrees of variability, their differences are rather subtly encoded in the whole-brain variational patterns that are distributed across its entire activity.”

Next, we moved our interpretation that was deleted from Introduction Sectionto Discussion Section(lines 178-205, in the current version of the manuscript). The newly added part reads as follows.

In this respect, our findings appeared to be more in line with Lindquist et al. [25] which found no single region that uniquely represented a specific affect (e.g., a region solely associated with the negative, neutral, or positive), thereby implying further evidence in favour of the affective workspace hypothesis [8] than the bipolarity [13] or the bivalent hypotheses [1416]. For instance, considering the bipolarity hypothesis [13] stance on attributing the positive and negative affect to the opposing ends of a single dimension [17,18], one may expect their respective brain activity to exhibit an anti-correlation. Contrary to this expectation, we observed that the brain variability associated with these affect to be positively correlated with one another. On the other hand, the emphasis by bivalent hypothesis [1416] on the presence of two distinct and independent brain systems for the positive and negative affect [1416] makes it plausible to expect that the observed brain activation that associates with these states to originate mostly from non-overlapping regions. Although our analyses identified a number of brain regions whose DEs significantly differed between negative, neutral, and positive affect, these regions were common among these affect states. Furthermore, we also observed that limiting the linear model’s feature space to solely include these regions resulted in significantly worsening of its performance. In this regards, the affective workspace hypothesis [8] argues that the positive and negative affect are the brain states that are supported by flexible and distributed brain regions [19]. Our findings appeared to be in line with this line of reasoning due to the following observations. First, we observed that the whole-brain variability in response to negative, neutral, and positive affect states were positively correlated. This indicated that any change in variational information of one affect (e.g., increase or decrease in the brain signal variability) can be explained in terms of a linear change in the variability of the other that is in the same direction as of the first one. This perspective found further evidence in the results of our simple linear model that was able to classify the negative, neutral, and positive affect with a significantly above average accuracy. Second, we also observed that the variational information of these affect states were distributed in the whole-brain activity and that their differences were quantified in terms of differing level of variability in these regions’ variability (i.e., information processing as quantified by their respective DEs) than their de/activation. However, future studies and analyses are necessary to draw a more informed conclusion on these observations.”

Reviewer’s Response: #4: Line 73: How many participants? Are these maps averages of all participants?

Author’s Response: There were15participants(7 males and 8 females; Mean (M) = 23.27, Standard Deviation (SD) = 2.37)in SEED dataset. This information can be found in Section5.1. The Dataset, lines 302-303in the current version of the manuscript.Out of these15 individuals,we excluded one participant since all of her/his EEG channels failed our data validation procedure (potentially due to excessive movement although we cannot confirm this), as explained in Section5.2. Data Selection and Validation, lines 349-352, in the current version of the manuscript (please also see our response to “Reviewer’s Comment: #22: Line 235: Bad channel removal and replacement ...”).

In Figure 1 (page 3in the current version of the manuscript), each spatial map corresponds to one individual (i.e., 14 maps, per affect, for 14 individuals). We updated the caption of Figure 1 to include this information as follows.

Figure 1. Spatial maps of participants’ whole-brain DE associated with (A) Negative (B) Neutral (C) Positive affect. Each of these maps corresponds to one individual that was included in the present study. For each individual, we first computed (for each channel separately) the average DE (per channel) of all movie clips’ DEs that were associated with a given affect. We then used these average DEs for each channel to construct these maps. Differential patterns of participants’ whole-brain variability in three different affect states is evident in these subplots.

With regards to how these DE averages were obtained, please refer to our response to “Reviewer’s Comment:#24: Lines 253 to 259: IMPORTANT.”

Reviewer’s Comment: #5: Line 76: figure 1 Letters B and C are not correct. Positive is C and neutral B, according to Figure 1 legend.

Author’s Response: We corrected this typo in the current version of the manuscript.

Reviewer’s Comment: #6: Line 82: Significant difference between negative, neutral and positive affect on what? Each electrode’s DE or whole brain DE?

Authors’ Response: It refers to the whole-brain DE. We modified this line (line88,in the current version of the manuscript) to reflect this fact as follows.

Kruskal-Wallis test identified a significant difference of the whole-brain DEs between negative, neutral, and positive affect ...

Reviewer’s Comment:#7: Line 82: How was this test performed? Why a non-parametric test?

Authors’ Response: We performed a pair-wise Wilcoxon test between every pair of affect (i.e., negative vs. positive, negative vs. neutral, and positive vs. neutral) on whole-brain DEs associated with these affect states. This information is provided in Section5.4.1. DE Analyses, lines 396-403in the current version of the manuscript.

With regardsto the use of non-parametric tests (i.e., Kruskal-Wallis and Wilcoxon), we checked our calculated DEs (per individual andper EEG channel / per individual andall EEG-channels / all individuals andper EEG channel (e.g., all F7s of all participants) / all individuals and all EEG channels) and found that they did not follow normal distribution.Therefore, we chose non-parametric tests. To clarify this, we added the following to the current version of the manuscript (Section DE Analyses, Lines 473-478):

With regard to our analyses, there are two points that are worth further clarification. They are the choice of non-parametric tests and the follow-up bootstrap test of significance. Prior to our analyses, we checked the calculated DEs of the participants in each of the negative, neutral, and positive affect states (separately as well as combined, with respect to the both individuals and the EEG channels for each of the affect) and found that they did not follow normal distribution. Therefore, we opted for non-parametric analyses.

Reviewer’s Comment: #8: Line 87: A test of significance of what?

Authors’ Response: We modified this sentence to read as follows (lines 95-97,in the current version of the manuscript).

Figure 3 shows the results of the paired two-sample bootstrap test of significance difference between DEs in negative, neutral, and positive affect (10,000 simulation runs) at 95.0% (i.e., p < .05) confidence interval (CI).”

Reviewer’s Comment: #9: Line 87, Figure 3, Table 1: Why do we need this paired bootstrap test? Shouldn’t the previous post-hoc Wilcoxon test be enough? Is it because it’s paired? If so, it should be briefly explained why this is relevant.

Authors’ Response: Our analyses were performed based on a small sample of participants (i.e., 14 individuals). Although, our non-parametric tests (please refer to our response to “Reviewer’s Comment: #7: Line 82: How was this test performed? Why a non-parametric test?” where we explained why we used non-parametric statistics)indicated significant difference between whole-brain DEs of these individuals in response to negative, neutral, and positive affect stimuli, we were concerned thatthese results might be due to only a subsample of individuals’ DEs (i.e., lack of central tendency and hence distorted data) than an observation that might be attributed to the effect of affect on whole-brain variability (i.e. within the scope of the present data). This concern was further strengthened by the result of the non-normality test of our data. Applying the bootstrap test allowed us to falsify this possibility. Specifically, by adapting thistest (i.e., random sampling with replacement) that was carried for 10,000 simulation runs (therefore allowing for potential outliers to be repeated with higher probability and more frequently) at 95% confidence interval (CI) (i.e., p < .05 significance level) enabled us to observe whether these results were indeed reliable, given the small sample in the present study: the CI made it possible to verify thatour results exhibited the central tendency and the overall test showed thatthedistribution of our test results approximated the normal distribution, thereby adhering the law of large number and the central limit theorem.

To further clarify the use of bootstrap test in our study, we added the following to the current version of manuscript (lines 478-485):

In the case of bootstrap, on the other hand, we realized that our analyses were performed based on a small sample of participants (i.e., fourteen individuals). Therefore, it was crucial to ensure that any significant results that we observed in our analyses were not due to a subsample of individuals’ DEs (i.e., distorted data and hence lack of central tendency). This concern was further strengthened by the result of the non-normality of DEs. Applying the bootstrap test (i.e., random sampling with replacement) that was carried out for 10,000 simulation runs (therefore allowing for potential outliers to be repeated with higher probability and more frequently) at 95% confidence interval (i.e., p < .05 significance level) enabled us to further verify our results.

Reviewer’s Comment: #10: Line 115: Please state size of training and holdout samples.

Authors’ Response:As adequately noted by the reviewer(comment #14), we used one participant as test and the remainder of the participant for training. We then repeated this process for each of the participant, thereby obtaining fourteen prediction results (i.e., one for each participant as test). In the case of test set, we included all negative, neutral, and positive affect data of the selected participant in the test set. To further clarify this point, we added the following to Section4.4.2. Linear Model Training Lines430-435:

This resulted in fourteen train-test runs (i.e., first participant used as test data and the remainder of participants used for training the model, Second participant used as test data and ..., fourteenth participant used as test data and the remainder of participants used for training the model). We then tested the model’s performance (i.e., per training scenario) on the holdout data. We repeated this procedure for every participant. This resulted in fourteen different test cases which we used to compute the linear model’s accuracy and confusion matrix.”

Reviewer’s Comment: #11: Line 115: what exactly is a “1-holdout setting”? (training and test set?)

Authors’ Response: Please refer to our response to Reviewer’s Comment “#10: Line 115: Please state size of training and holdout samples.”

Reviewer’s Comment: #12: Line 116: Does this mean there was a leave one out cross validation with the training sample? Not clear.

Authors’ Response: Please refer to our response to Reviewer’s Comment “#10: Line 115: Please state size of training and holdout samples.”

Reviewer’s Comment:#13: Figure 5: confusion matrices could benefit from a color scale (blue for the diagonal, red for the off-diagonal items, light for low values, stronger for higher values). Why only the panels C and D are called confusion matrices? Aren’t A and B also? Increase font size of panel E…

Authors’ Response:We color-coded the confusion matrices and adjusted the font size of panel E (Figure 6, panel C, page 7,in the current version of the manuscript). We also realized that the use of subplots B and D were redundant and might confuse the reader. Therefore, we removed these subplots from this figure and modified its caption as follows.

Figure 6. Linear model’s confusion matrices associated with its prediction accuracy in negative, neutral, and positive affect in 1-holdout setting using (A) whole-brain DEs and (B) subset of channels with the similar pattern of significant differences as whole-brain DEs. These results show the model’s performance after data from every participant was used for testing. Correct predictions, per affect, are the diagonal entries (blue) of these tables and the off-diagonal entries (red) show the percentage of each of affect that was misclassified (e.g., positive affect misclassified as negative affect). (C) Two-sample bootstrap test of significance (10,000 simulation runs) at 99.0% (i.e., p < .01) confidence interval between the accuracy of linear model using the whole-brain DEs versus subset of channels with the similar pattern of significant differences as whole-brain DEs.”

To provide further insight into thelinear model’s performance, we addeda new table (Table 3, page 8,in the current version of the manuscript) that summarizedthe precision, recall, and F1-score of the model.

To reflect these changes in the text, we modified the manuscript’s content (lines123-151in the current version of the manuscript) as follows:

Figure6(A) shows the linear model’s prediction accuracy in negative, neutral, and positive affect in 1-holdout setting using whole-brain DEs. In this setting, we considered a single participant’s positive, negative, and neutral affect data as test set and used the remaining participants’ data for training the linear model. We then tested the model’s performance on the holdout test data. We repeated this procedure for every participant. Subplot (A) in Figure 6 indicates that the use of whole-brain DEs to quantify the brain variability of the participants enabled the linear model to predict their negative, neutral, and positive affect with significantly above chance level (chance level accuracy ≈ 33.33%, given the three-affect settings). One-sample bootstrap test of significance (10,000 simulation runs) at 95.0% (i.e., p < .05) confidence interval verified this observation (M = 78.52, SD = 6.45, CI95% = [61.90 88.10]). We also observed that these predictions were substantially higher in the case of negative and neutral than the positive affect. An inspection of Figure 6 (A) reveals that a relatively larger number of participants’ positive affect was misclassified as negative affect by the linear model. Table 3, top row, summarizes the precision, recall, and F1-score associated with the linear model’s performance in negative, neutral, and positive affect in 1-holdout setting using whole-brain DEs.

We further checked the linear model’s performance using the subset of channels with the similar pattern of significant differences as whole-brain DEs (Figure 6(B)). Our analyses (for details, see Appendix A) identified that these channels were in the frontal (F3, FZ, F4), frontocentral (FCZ), central (C5, C3, C1, C2, C4), centroparietal (CP3, CP1, CP2), parietal (P3, POZ), and occipital (CB1, OZ) regions. Figure 6 (B) corresponds to the 1-holdout setting using this subset of participants’ channels. This subplot reveals a substantial reduction of linear model’s accuracy in comparison with the case in which we used the whole-brain DEs. Wilcoxon test identified a significant difference between the linear model’s accuracy in these two settings (p <.001, W(998) = 7.23, r = .23) which we furtherverified using two-sample bootstrap test of significance (10,000 simulation runs) at 99.0% (i.e., p < .01) confidence interval (Figure 6 (C)) (Mdi f f erence = 12.54, SDdi f f erence = 1.63, CIdi f f erence = [8.27 16.67]) (see Appendix C for randomized 500 simulation runs of these results). Table 3, bottom row, summarizes the precision, recall, and F1-score associated with the linear model’s performance in negative, neutral, and positive affect in 1-holdout setting using this subset of channels with the similar pattern of significant differences as whole-brain DEs.”

We applied a similar modification to Figure A4. (i.e., the randomized settingpage23,in the current version of the manuscript). It reads as follows.

Figure A4. Confusion matrices associated with randomized affect prediction based on individual’s brain variability by the linear model in 1-holdout setting (500 simulation runs, per affect) using (A) whole-brain and (B) subset of channels with the similar pattern of significant differences as whole-brain DEs. These results show the model’s performance after data from every participant was used for testing. Model’s accuracy for correct and incorrect predictions are the diagonal and off-diagonal entires.

We also added a new table (Table A4., page 23,in the current version of the manuscript) that summarizedthe linear model’s precision, recall, and F1-score associated with this randomized setting. To reflect these changes, we modified its content (Appendix C Linear Model’s Prediction - Randomized Case, Lines 414-434) as follows:

FigureA4shows the linear model’s prediction of negative, neutral, and positive affect in a randomized setting (for both training scenarios) in which we first randomly selected an individual and then picked, at random, only one of this selected individual’s negative, neutral, or positive affect data for 1-holdout testing. We then used the remaining data that did not include any of the randomly selected individual’s affect data for training. To guarantee an unbiased estimate of the linear model’s prediction, we continued this random selection until every affect was selected 500 times. This differed from the first 1-holdout setting (reported in the main manuscript) in which we used data pertinent to all three affect of the selected individual for 1-holdout test.

The confusion matrix associated with these 500 rounds of individual’s random selection for 1-holdout test is shown in Figure A4(A). Similar to the case of single-trial, per participant (Section 6.2 in the main manuscript), the use of whole-brain variability enabled the linear model to predict the negative, neutral, and positive affect with the accuracy that was significantly above chance level (chance level accuracy ≈ 33.33%, given the three-affect setting). In addition, we also observed that these predictions were substantially higher in the case of negative and neutral than the positive affect. An inspection of Figure A4(B) reveals that a relatively large number of participants’ positive affect was (similar to the case of single-trial, per participant; Section 6.2 in the main manuscript) misclassified as representing their negative affect (i.e., 141 out of 183 misclassified positive state in Figure A4 (B) bottom row entry).

TableA4 summarizes the precision, recall, and F1-score of the linear model’s performance associated with this randomized setting. A comparison between entries of this table and Table 3 reveals that these model’s metrics remained stable.

Reviewer’s Comment: #14: 118: So this is a leave one out cross validation?

Authors’ Response: Please refer to our response to Reviewer’s Comment “#10: Line 115: Please state size of training and holdout samples.”

Reviewer’s Comment: #15: Line 118-119: So, in line 114 you said “Figure 5 (A) shows the linear model’s prediction accuracy in negative, neutral, and positive affect in 1-holdout setting using whole-brain DEs”. This was for one participant only? Then in these lines you say “he confusion matrix associated with the 1-holdout setting in the case of whole-brain DEs is shown in Figure 5 (B)”. The difference is that panel B is for the entire leave one out cross validation? This isn’t totally clear and the readers need to know from which samples these accuracies are computed from, the size of the test samples, etc…

Authors’ Response:Please refer to our response to Reviewer’s Comment: #13: Figure 5: confusion matrices could benefit from a color scale...”

Reviewer’s Comment: #16: Line 121 to 128: Why a bootstrap and not a p-value obtained from a binomial distribution?

Authors’ Response:Given the small number of sample (i.e., fourteen participants), the total number of accuracies of themodel was too small (i.e., fourteen runs) for any test of significant (parametric or non-parametric) to be meaningful. Therefore, weadapted the one-samplebootstrap(please also refer to our response to “Reviewer’s Comment: #9: Line 87, Figure 3, Table 1: Why do we need this paired bootstrap test?” for a broader explanation of the use of bootstrap, in general, in our study)test in which we tested therobustness of the accuracyof the model byrandomly sampling(with replacement) fourteen accuracies for 10,000 runs. This procedure allowed us to verify whether thesepredictions were significantly above chance levelin a long run (i.e., 10,000 times of their resampling).Considering the fact that we used 99% confidence interval while performing this test, itsresult indeed correspondedto the p-valuep < .01significancelevel.

To further verify this point, we added the following to Section5.4.2. Linear Model Training, lines435-432:

Given the small number of sample (i.e., fourteen participants), the total number of accuracies of the model was too small (i.e., fourteen runs) for any test of significant (parametric or non-parametric) to be meaningful. Therefore, we adapted a one-sample bootstrap test in which we tested the robustness of the model’s accuracies during the fourteen 1-holdout runs. For this test, we performed a one-sample bootstrap test of significance (10,000 simulation runs) at 99% confidence interval (CI) to check whether the model’s accuracy was above chance level (i.e., ≈ 33.33%). Considering the fact that we used 95% confidence interval while performing this test, its result indeed corresponded to the p-value p < .01 significance level.”

Reviewer’sComment:#17: Line 212: No linear drift removal?

Authors’ Response:We did not notice any mention of it in articles [59,60] as well as the website associated with this dataset repository.

Reviewer’sComment:#18: Figure 6. Panel A) “seconds” is not well written. Are the subjective self-reported emotions associated with DE? Also, are they associated with the supposed elicited emotions?

Authors’ Response: We modified the Panel A in this figure’scaption (Figure 7, page12,in the current version of the manuscript) to further explain these self-assessed responses (please also refer to our response toReviewer’s Comment: #1 Line 11: “Our results suggest that the whole-brain variability significantly differentiates between the negative, neutral, and positive affect”. This is no accurate…. for explanations about ranking of the movie clips, use of self-assessed responses, etc. )

Figure 7. (A) Schematic diagram of an experiment as described in [59]. Each experiment included a total of fifteen movie clips (i.e., n = 15), per participant. In this setting, each movie clip was proceeded with a five-second hint to prepare the participants for its start. This was then followed by a four-minute movie clip. At the end of each movie clip, the participants were asked to answer three questions that followed the Philippot [61]. These questions were the type of emotion that the participants actually felt while watching the movie clips, whether they watched the original movies from which the clips were taken, and whether they understood the content of those clips. The participants responded to these three questions by scoring them in the scale of 1 to 5.”

Reviewer’s Comment: #19: Line 224: state that length is data points and not seconds (even though it’s obvious).

Authors’ Response:We modified this sentence as follows (lines 340-342, in the current version of the manuscript).

To ensure that the DE computation was not affected by the length of EEG channels, we first ensured that all EEG recordings that were included in our study were sufficiently long in term of number of their respective data points …

Reviewer’s Comment: #20: Line 226: Two lines before the minimum number of observed data points was 37001 and not 37000…

Authors’ Response:We corrected this typo.

Reviewer’s Comment:#21: Line 232: Given the low N, why was this participant promptly excluded? Couldn’t the data pre-processed? Detect where the unit root is; apply a derivative to remove unit root, etc…

Authors’ Response:We had to exclude this participant since all her/his EEG channels failed during our data validation step, as explained in Section 5.2. Data Selection and Validation.Furthermore, we observed this failure in all of the participant’s experimental sessions. Although we could not verify this, we believethat it might havebeendue to theexcessive movement bythisparticipant. Therefore, we tried to remove such a potential effect using a number of approaches including the use of PCA, ICA, exponential moving average, and Kalman filter [2,3,4]. However, the problem with thedata persistedanddid not pass our data validation test. We also tried the unit root removal unfortunately with no success. Therefore, we decided to exclude this participant.

With regard to small sample size, we also added a new paragraph in the newly added Section 4.Limitations and Future Direction to highlight the need for further study with a larger sample which reads as follows (lines 295-299, in the current version of the manuscript).

Last, although neuropsychological findings indicate that the individuals’ ability to experience pleasant/unpleasant feelings to express these subjective mental states in terms of such attributes as positive or negative [25] to be the unifying and common concept across cultures [11,12], future research that includes a larger human sample as well as different age groups along with more cultural diversity is necessary for drawing a more informed conclusion on the findings that were presented in this article.”

[2]Cooper, R., Selb, J., Gagnon, L., Phillip, D., Schytz, H.W., Iversen, H.K., Ashina, M. & Boas, D.A., A systematic comparison of motion artifact correction techniques for functional near-infrared spectroscopy, Frontiers in Neuroscience, 6 , 147 (2012).

[3]Sun, L. and Hinrichs, H., 2009. Simultaneously recorded EEG–fMRI: Removal of gradient artifacts by subtraction of head movement related average artifact waveforms. Human brain mapping,30(10), pp.3361-3377.

[4]Griffanti, L., Salimi-Khorshidi, G., Beckmann, C.F., Auerbach, E.J., Douaud, G., Sexton, C.E., Zsoldos, E., Ebmeier, K.P., Filippini, N., Mackay, C.E. and Moeller, S., 2014. ICA-based artefact removal and accelerated fMRI acquisition for improved resting state network imaging.Neuroimage,95, pp.232-247.

Reviewer’s Comment:#22: Line 235: Bad channel removal and replacement (by neighbors average for example) is common practice in some EEG analysis. Or another procedure to remove unit roots. Why isn’t it applied here? Replacing an entire session with the subsequent (second or third) seems to add more bias than removing the channel. Could you explain why this was done?

Authors’ Response: The main goal of our study was to examine the whole-brain variability (as it was quantified by theentropy of multiple EEG channels’ recordings) to negative, neutral, and positive affect.To this end and given the low spatial resolution of EEG recordings, we did not consider reducing the channels by removing the ones that were not passing our data validation step as an option.Using all channels provided us with a relatively good spatial resolution of the brain activity in response to different affect states, thereby complementing the high temporal resolution of EEG. On the other hand, we did not consider their replacement with, for example, average of their neighboring channels (or any other transformation of these neighbors) due to the potential increase in spurious and/or redundant results. Infact,neuroscientific findings identify a functional interactivity between the brain regions (reference [56] in the current version of the manuscript) in which signals from individual cortical neurons are shared across multiple areas and thus concurrently contribute to multiple functional pathways (reference [86] in the current version of the manuscript). Using such alternatives as replacing the affected channels with some transformation of their neighboring channels (e.g., their average) could have resulted in unreliable observations by adding (although superficially) to such information flow/sharing (references [87,88] in the current version of the manuscript). We reflected these concerns in the newly added Section4. Limitations and Future Directionas follows (lines 265-273, in the current version of the manuscript).

It is also crucial to note that in the present study we utilized the entropy (i.e., DE) of EEG time series of human subjects for analysis of the whole-brain variability in response to negative, neutral, and positive affect states. In this regard, research identifies a functional interactivity between the brain regions [56] in which signals from individual cortical neurons are shared across multiple areas and thus concurrently contribute to multiple functional pathways [89]. However, our analyses did not take into consideration the potential effect of such flow of information among different brain regions in response to differential affect states while computing the EEG channels’ DEs. Therefore, future research to study the possibility of such potential information flows [90,91] is necessary to better realize the utility of the brain variability in emergence of differential affect states in response to natural stimuli.”

Given the above two observations and considering the fact that the data included in our study was already consisted of small sample, we did not want to lose additional data. Therefore, we used the data from other sessions of the two participants in which two channels of their EEG channels did not pass our data validation. In this dataset,all participants used the same 15 movie clips that were presented to them in the exact sameorder during all the sessions that these individuals participated in.Furthermore, all the data that are included in thisdataset were selected based on the same affect-elicitation validation procedure (please refer to our response toReviewer’s Comment: #1 Line 11: “Our results suggest that the whole-brain variability significantly differentiates between the negative, neutral, and positive affect”. This is no accurate… for further details).Given these observations, it was plausible to consider data of a participant that was recorded in one session and in response to a given affect (e.g., negative)to be a goodrepresentative of her/his brain activity in another session and to the same affect state (i.e., negative in the case of this example)that was induced by the same movie clips. In other words,assuming no substantial change in individuals’ brain functioning, it is plausible to expect that their brain not to respond significantly differently to the same stimuli that was presented to them in different days.However, we reflected on the potential benefits and use-cases of these other data of the participants in the in the newly added Section4. Limitations and Future Directionas follows (lines274-294,in the current version of the manuscript).

The main goal of the present study was to verify whether the underlying brain substrates for negative, neutral, and positive affect were rather subtly encoded in the whole-brain variational patterns that were distributed across its entire activity. As a result, we primarily focused on the statistical analyses of the whole-brain variability in terms of its distributed information processing (i.e., its entropy) and used the linear model as a supportive evidence that showed that whole-brain variability in fact resulted in better representation of these three affect states than the use of selected brain regions that also showed similar statistically significant differences. In this respect, including several measurements of the same subjects that watched the same movie clips in the same order in multiple days would have only complicated the interpretation of the results due to such issues as multiple-comparison as well as potentially confounding factors such as redundant information (e.g., assuming no substantial change in individuals’ brain functioning, it is plausible to expect thattheir brain not to respond significantly differently to the same stimuli that was presented to them in different days). Therefore, we decided to reduce the possibility of occurrence of such issues and confounders by only including one out of three sessions of each of the participants. However, such multiple measurements can benefit the future research by providing an opportunity to test for the reproducibility of the current results. For instance, one can compute their respective whole-brain variability using their respective DEs and compare their corresponding neural substrates with the results in the present study. They can also be utilized as one larger test set (i.e., all together) to verify whether the linear model in this study can preserve its accuracy on this new data. The latter scenario can become a valuable testbed for the cases in which training personalized models for the individuals is desirable (e.g., personalized socially-assistive robots [92]).”

Reviewer’s Comment:#23: Line 241: were self-reported ratings significantly different between (and matching) the different emotional clips? It’s important that the self-reported emotions match the affect states that the clips are supposed to elicit.

Authors’ Response:Please refer to our response to “Reviewer’s Comment: #1 Line 11: “Our results suggest that the whole-brain variability significantly differentiates between the negative, neutral, and positive affect”. This is no accurate…”

Reviewer’s Comment: #24: Lines 253 to 259: IMPORTANT.This is the most relevant point regarding this paper and if not approached properly, the paper shouldn’t be published, in this reviewer’s opinion. It is clear from the description that the DEs are first averaged by condition (i.e. affective state) which results in a [1x 62 x affect x participant] matrix as stated in lines 258 and 259. The problem is that the order of averaging matters, as has been demonstrated by a response to a result of a recent BCI controversial study (https://doi.org/10.1371/journal.pbio.2004750). Here, the author shows that averaging first by trials and then by channels removes the variance over trials resulting in very low variance between channels which will increase the chance of finding a false significant difference. Furthermore the author of the cited work writes:

This does not mean that averages should not be used for statistical analysis, but that the order of averaging matters. If the data are first averaged over channels and then over trials, the variance over sessions is retained.

And:

Regarding the averaging over channels, it should also be mentioned that averaging across channels is generally not recommended. Although it is not incorrect, averaging channels can reduce or even cancel out an effect in the data if the signal of interest is highly localized or shows different patterns across different brain areas. For this reason, an analysis treating each channel independently (as in S1 Text) is recommended.

Given these points, and that the main results of the paper under review are on the averaged DEs over channels, I have to ask the authors to critically explain if and why their methodology falls within or outside the described problem. I also have to ask the editor to take this comment with high relevance since we shouldn’t allow results stemming from already known incorrect methodology to be published. I should also state that this can be a consequence from this format of relegating the methods to the end, instead of having them before the Result section. The Methods section should be as relevant as the Results section in journals heavy on methodological work.

Also, it’s not stated here how the whole-brain DE is computed So I’m assuming an average over the affect dimension of the [1 x 62 x affect x participant] matrix.

Authors’ Response: We fully agree with the reviewer’s comment and appreciate the reviewer’s observation. However, we believe that the reviewer’s adequate comment has been a misunderstanding that was due to thelack of clear explanation by the authors while describing the process of DE computation and their subsequent averaging.

The two excerpts of the cited reference by the reviewer (i.e.,https://doi.org/10.1371/journal.pbio.2004750) highlight the pitfall of averaging over the channels while quantifying the effect of interest (in our case, for example, computing DEs). However, we did not follow such a procedure.

Every affect in the dataset that we used in our studywas associated with 5 movie clips (i.e., 15 movie clips in total). For each of these trail, there was an associated 62-channel EEG recordings that are in their own respective separate files. In other words, there are 5 separate files of 62-channel EEG recordings for each of the negative, neutral, and positive affect states. Given these data organization, we applied the following steps to calculate the DEs:

For a given affect (e.g., negative) and for each participant (i.e., person-by-person), we accessed the associated 5 files of 62-channel EEG recordings one-by-one.

For each of these files (per affect, per participant), we then utilizedeach of the EEG channels one-by-one and computed its DE. This resulted in 62 DE values (i.e., one per EEG channel)for each of the 5 movie clips of the given affect.

At the end of steps 1 & 2, we had 5 separate sets (i.e., one for each movie clip in each affect setting and for each participant) of 62 DEs (i.e., one DE for each of the 62 EEG channels).

We used these 5 separate sets of 62 DEs and computed the average DE for each channel. For example, for channel F7 in a given affect state (e.g., negative) and for each participant, we had five DEs (i.e., one DE for each movie clip:DEF7movie clip1 , DEF7movie clip2 , DEF7movie clip3 , DEF7movie clip4 , DEF7movie clip5) that were computed in steps 1 & 2. We averaged these five DEs i.e., mean([DEF7movie clip1 , DEF7movie clip2 , DEF7movie clip3 , DEF7movie clip4 , DEF7movie clip5]).It can be appreciated that any permutation/orderingof these five values have no effect on the result of computed average DE for F7 in a given affect state (e.g., negative affect).

We repeated these steps for each affect and each participant.

However, the reviewer’s comment made us realize that our explanation fell short in adequately describing the procedure we adapted while computing the DEs. In the current version of the manuscript, we therefore modified the content of Section5.3. DE Computations, lines358-373, as follows.

We computed DE using the full-length EEG data (i.e., 37000 data points) per channel, per participant, per affect, and adapted the following procedure for its computation. For a given affect (e.g., negative), we accessed each individuals’ associated five files (i.e., one file for each of the movie clips, per affect) of 62-channel EEG recordings one-by-one. For each of these files, we then used each of the EEG channels one-by-one and computed its DE (i.e., one DE for each of the EEG channels). This resulted in five separate sets (i.e., one set for each of the movie clips) of sixty-two DEs (i.e., one DE for each of the EEG channels). Next, we utilized these five separate sets of 62 DEs and computed the average DE for each channel. For example, for channel F7 (Figure 7(B)), we had five DEs (i.e., one DE for each of the movie clips of a given affect (e.g., negative affect): [DEF7movie clip1 , DEF7,movie clip2 , DEF7movie clip3 , DEF7movie clip4 , DEF7movie clip5].We averaged these five DEs i.e., mean([DEF7movie clip1 , DEF7movie clip2 , DEF7movie clip3 , DEF7movie clip4 , DEF7movie clip5]that the ordering of these values have no effect on the computed average DE for F7 in a given affect state (e.g., negative affect) [69]. We repeated this procedure for each participant (i.e., fourteen) and each affect (i.e., negative, neutral, and positive), thereby computing their averaged brain variability in response to a given affect. This process resulted in 1 × 62 vectors, per affect, per participant, where 62 refers to the number of EEG channels (Figure 7(B)).

We used a non-parametric DE estimator by Kozachenko and Leonenko [64] that estimates the differential entropy of a continuous random variable using nearest neighbour distance [65]. It is worthy of note that a number of previous studies has adapted DE for emotion classification [66,67].”

Reviewer’s Comment: #25: Line 167: replace “comprised of” with “comprises”

Authors’ Response:We did not find “comprised of” at line 167 which could be due to some formatting changes that were applied by the editorial office to the manuscript. However, we checked all occurrences of the term in our manuscript (two in total) and changed them.

Reviewer’s Comment: #26: Line 267: Why non-parametric?

Authors’ Response:Please refer to our response to Reviewer’s Comment: #9: Line 87, Figure 3, Table 1: Why do we need this paired bootstrap test?”and Reviewer’s Comment: #16: Line 121 to 128: Why a bootstrap and not a p-value obtained from a binomial distribution?”

Reviewer’s Comment:#27: Line 269: Why the further validation?

Authors’ Response:Please refer to our response toReviewer’s Comment: #9: Line 87, Figure 3, Table 1: Why do we need this paired bootstrap test?”andReviewer’s Comment: #16: Line 121 to 128: Why a bootstrap and not a p-value obtained from a binomial distribution?”

Round 2

Reviewer 3 Report

Thank you for your revised manuscript. Most points were addressed and comment #24 is now solved.

However, there are still some points I would like to bring to your attention. I will add them in bold, following your responses in italic.

Reviewer’s Comment: #24: Lines 253 to 259: IMPORTANT.

Thank you for your reply. This comment was addressed.

Your response:

How elicited affect of the movie clips were verified: To ensure that the selected movie clips indeed elicited the targeted affect, a preliminary study [60] was conducted where twenty participants were asked to assess a pool of movie clipsin afive-point scale.

Comment #R1

Were these the same participants? This study is from 2015. How can we be sure that because it elicited emotons in 20 participants 4 years ago, on another country, it I will elicit the same emotions in 15 participants now?

Can you run an ANOVA on the self-reported emotions and see if the movie type has an effect on these self-reports?

I understand that there might be no correlation between self-report emotion and actual emotion. But if the movies are only validated with 20 subjects, we need to know there is actually a maniputalion happening in this study that the DEs is picking up on. Otherwise you cannot state that it positive, neutral and negative affect…

Reviewer’s Response: #4: Line 73: How many participants? Are these maps averages of all participants?

Author’s Response: There were15participants(7 males and 8 females; Mean (M) = 23.27, Standard Deviation (SD) = 2.37)in SEED dataset.

Comment R2: How was this number obtained? Is there a power analysis or prior study that justifies the use of N=15?

For more information about best practices for EEG / MEG experiments the COBIDAS document can be a useful guideline:
https://www.humanbrainmapping.org/i4a/pages/index.cfm?pageid=3728

Reviewer’s Comment: #10: Line 115: Please state size of training and holdout samples.

Authors’ Response:As adequately noted by the reviewer(comment #14), we used one participant as test and the remainder of the participant for training. We then repeated this process for each of the participant, thereby obtaining fourteen prediction results (i.e., one for each participant as test). In the case of test set, we included all negative, neutral, and positive affect data of the selected participant in the test set. To further clarify this point, we added the following to Section4.4.2. Linear Model Training Lines430-435:

Comment R3: This is a very uncommonly low number of training samples. The implications of this low N should be discussed, by at least citing other studies where such a low number was used.

Reviewer’s Comment:#13: Figure 5: confusion matrices could benefit from a color scale (blue for the diagonal, red for the off-diagonal items, light for low values, stronger for higher values)

Comment R4: This was still not done. The idea was that higher amplitude values would have stronger color and lowe ramplitude values lighter color (and negative could be blue and positive white). See example: https://ch.mathworks.com/help/stats/confusionmat.html

Reviewer’sComment:#18: Figure 6. Panel A) “seconds” is not well written

Comment R5: “Seconds” is still written as “secends”

Reviewer’s Comment:#23: Line 241: were self-reported ratings significantly different between (and matching) the different emotional clips? It’s important that the self-reported emotions match the affect states that the clips are supposed to elicit.

Comment R6: It’s still not clear if the elicited emotions are different between clips.

Comment R7: Is this whole brain signal difference also present using frequency band analysis? Or is entropy the best tool for this task?

Author Response

First and foremost, the authors would like to take this opportunity to thank the reviewer and the associate editor for their time and kind consideration to review our manuscript. The comments by the reviewer enabled us to further improve the quality of our results and their presentation substantially.

In what follows, we provide point-by-point responses to the second-round comments and concerns raised by the reviewer 3.

Sincerely,

Reviewer 3

Reviewer’s Comment: Comment #R1 Were these the same participants? This study is from 2015. How can we be sure that because it elicited emotons in 20 participants 4 years ago, on another country, it I will elicit the same emotions in 15 participants now?

Can you run an ANOVA on the self-reported emotions and see if the movie type has an effect on these self-reports? I understand that there might be no correlation between self-report emotion and actual emotion. But if the movies are only validated with 20 subjects, we need to know there is actually a maniputalion happening in this study that the DEs is picking up on. Otherwise you cannot state that it positive, neutral and negative affect…

Author’s Response: We believe that this comment is due to a misunderstanding. We did not conduct a new experiment but used a publicly available dataset called SEED (i.e., reference [59] in the current version of the manuscript and also Section 1. Introduction, lines 60-64,Section 3. Discussion, lines 153-156, and Section 5.1. The Dataset). We also included information about the priorstudy (i.e., reference [61] in the current version of the manuscript)that detailedthe process through which the movie clips that were used in SEED as well wereselected and evaluated. All the individuals in these studies (including the 20 volunteers) were Chinese nationals.In addition, the two studies were not 4 years apart but both took place in 2015. The steps that led to publicly available SEED dataset that we also used in our study wereas follow.

Prior to [61], 20 volunteers took part in rating a pool of movie clips in a five-point scale. Based on this study, fifteen movie clips whose average score were ≥ 3 and ranked in the top 5 in each affect category were chosen. This resulted in 5 movie clips per negative, neutral, and positive affect.

Then in [61], the authors conducted an experiment that included 9 participants and verified that the movie clips were elicited the targeted affect (please also see our response to Reviewer’s Comment: Comment R3: This is a very uncommonly low number of training samples…).

Finally, they used these set of 15 movie clips whose selection and evaluation steps were outlined in steps 1 & 2 to conduct the study that was reported in [59]. The result of the latter study (i.e., [59]) was the publicly available SEED dataset. In [59], they also showed that these movie clips indeed induced the targeted affect in 15 participants that took part in SEEDexperiment and that they were able to classify these affect states using the brain activity of these 15 participants.

In summary, the 20 volunteers who initially rated (using the scoring criteria mentioned above)these movie clips were not part of the experimentsthat werereported in [59] and [61]. They were used as initial independent evaluators to obtain the most relevant movie clips from a pool of such clips that best represented the targeted affect (i.e., negative, neutral, and positive). This step resulted in selecting 15 movie clips (i.e.,5 movie clips per negative, neutral, and positive affect).Theynextused theseselected 15movie clips and validated themin [61] that included 9 separate participants (i.e., different from the original 20 participants). This showed that the movie clips that were labeled as eliciting the negative, neutral, and positive affectby initial independent 20 volunteerswere indeed able to induce such affect on the brain activity of a new set of participants (i.e., 9 participants in [61])that did not take part in theoriginal ratings of those movie clips. The same movie clips were then used in [59] (i.e., publicly available SEED dataset that weused in our study). Furthermore,[59] and [61] were conducted in 2015 andboth included the use of brain responses to the same movie clips that were labeled by another group of20 volunteers. In total, the entire study that led to generation of SEEDincluded 20 + 9 + 15 = 44 individuals.

We would also like to bring to the reviewer’s kind attention that [59] and [61] did not report the self-assessment responses. Furthermore, the self-assessment of the participants were not included in the publicly available SEED dataset. They were primarily used to choose a right set of movie clips for the purpose of generating the SEED results and its accompanying dataset.

To make this procedure more clear in our manuscript, we added the following paragraph to Section 5.1. The Dataset, lines 318-324,in the current version of the manuscript:

Prior to SEED experiment, the authors asked twenty volunteers to assess a pool of movie clips in a five-point scale based on which the fifteen movie clips (i.e., five clips per negative, neutral, and positive affect) whose average score were ≥ 3 and ranked in the top 5 in each affect category, were chosen. The authors further verified that the selected movie clips indeed elicited the targeted affect in a follow-up study [61] that included nine separate individuals who were different from the twenty volunteers that originally involved in rating and selection process of fifteen movie clips). The authors then used these movie clips in SEED experiment [59].”

With regard to verification of our results on differential effect of the negative, neutral, and positive affect, our results were in line with the findings in the literature on their effect, thereby extending these previous findings. For instance, we observed that the negative and the positive affect were associatedwith higher DE than the neutral state which was in accord with quantitatively higher information processing in emotional than neutral contexts [74] as well as the findings on increased brain activity with attention [75,76]. The higher DE in the negative than the positive affect also hinted at the effect of negative emotions on cortical activity [79,80]. We discussed this information in Section 3. Discussion, lines 166-172, in the current version of the manuscript.

In addition, our results on the bilateral whole-brain variability were in line with the findings on the brain functioning during story comprehension that identified such bilateral activations [86,87]. This extended the results on the occurrence of such a bilateral brain activity during auditory story comprehension to the case of movies (i.e., visual & auditory stories) as it has also been reported by Hasson et al. [88] and Jääskeläinen et al. [89]. We included this information in Section 3. Discussion, lines 219-225, in the current version of the manuscript.

Reviewer’s Comment: Comment R2: How was this number obtained? Is there a power analysis or prior study that justifies the use of N=15? For more information about best practices for EEG / MEG experiments the COBIDAS document can be a useful guideline:
https://www.humanbrainmapping.org/i4a/pages/index.cfm?pageid=3728

Authors’ Response: We did not find any referencetoCOBIDAS in SEED documentation.On the other hand, they reported that they selected their 15 participants based on the outcome of Eysenck Personality Questionnaire (EPQ) [60] personality traits. EPQ evaluates the individuals’ personality along three independent dimensions of temperament: Extraversion/Introversion, Neuroticism/ Stability, and Psychoticism/Socialisation. Eysenck et al. [60] reported that it appears that not every individual can elicit specific emotions immediately (even in the presence of explicit stimuli) and that individuals who are extraverted and have stable moods tend to elicit the right emotions throughout the emotion-based experiments. Therefore, the authors of SEED [59]adapted the same personality criteria that was reported by Eysenck et al. [60] (i.e., high ratesof extraversion and mood-stability based on EPQ outcome) to select the 15 individuals that participated in SEED experiment.

To further clarify on this point, we included the following to Section 5.1. The Dataset, lines 310-317:

The authors reported that these individuals were selected based on the Eysenck Personality Questionnaire (EPQ) [60] personality traits. EPQ evaluates the individuals’ personality along three independent dimensions of temperament: Extraversion/Introversion, Neuroticism/ Stability, and Psychoticism/Socialisation. Eysenck et al. [60] reported that it appears that not every individual can elicit specific emotions immediately (even in the presence of explicit stimuli) and that individuals who are extraverted and have stable moods tend to elicit the right emotions throughout the emotion-based experiments. Therefore, the authors adapted the same personality criteria that was reported by Eysenck et al. [60] to select the fifteen individuals that participated in SEED experiment.

Reviewer’s Comment: Comment R3: This is a very uncommonly low number of training samples. The implications of this low N should be discussed, by at least citing other studies where such a low number was used.

Authors’ Response: This is indeed an important issue that is quite pervasive in neuroscientific studies which can be due to the difficulty associated with conducting such studies (e.g., tedious and time-consuming procedure for recruiting human subjects that best suites the experimental paradigm, additional documentation and permission requirement associated with studies that involve the use of human subjects and other animals, finding the targeted age-range and/or gender-balance, etc.). In fact many studies draw their observations using a small number of participants. For instance, [94] included 18 participants who watched series of emotional images. Similarly, in [95] 18 participants watched a 25-minute movie clip, in [96] 13 participants watched colored images,and [97] included 9 participants only.Furthermore, many of these studies rely on individuals who mostly share the same geographical/cultural background. As an exception, our overview showed that [89] included 15 participants, three of whom were discarded due to the issues related to technical reasons/poor data quality (i.e., in total 12 participants were included in [89]). Out of these 12 participants, 11 were native Finnish speaker and 1 only was a native English speaker.

In fact, the comprehensive meta-analysis that was published in 2016 about the neural correlates of the affect (i.e., reference [25] in the current version of the manuscript) included 914 experimental contrasts that accounted for 6827 participants over the period 1993-2011 (i.e., 6827/914 ≈ 7.47 participants on average).

However, we also appreciateand agree with the reviewer’scomment on the importance of this issue. Therefore, we modified the last paragraph in Section 4. Limitations and Future Direction, lines 295-304, to more clearly emphasize on this point. The modified paragraph reads as follows.

Many neuroscientific studies are based on a small number participants [9497]. For instance, the recent comprehensive meta-analysis by Lindquist [25] that reported on an extensive coverage of 914 experimental contrasts in affect-related studies accounted for 6827 participants (i.e., 6827/914 ≈ 7.47 participants on average). Furthermore, most of the individuals that are included in these studies share the same geographical and/or cultural background (but also see [89] for a small deviation). Although neuropsychological findings indicate that the individuals’ ability to experience pleasant/unpleasant feelings to express these subjective mental states in terms of such attributes as positive or negative to be the unifying and common concept across cultures [11,12], future research that includes a larger human sample as well as different age groups along with more cultural diversity is necessary for drawing a more informed conclusion on the findings that were presented in this article.”

Reviewer’s Comment: Comment R4: This was still not done. The idea was that higher amplitude values would have stronger color and lowe ramplitude values lighter color (and negative could be blue and positive white). See example: https://ch.mathworks.com/help/stats/confusionmat.html

Authors’ Response: We apologize for missing on the better visualization of these subplots. In the current version of the manuscript (Figure 6, page 7 and Appendix A, Figure A4, page 23)we used Python seaborn to generate these confusion matrices.

Reviewer’s Comment:Comment R5: “Seconds” is still written as “secends”

Authors’ Response: We apologize for missing the misspelled “seconds” in the Figure 7 (A). It was corrected in the current version of the manuscript.

Reviewer’s Comment: Comment R6: It’s still not clear if the elicited emotions are different between clips.

Authors’ Response: Please refer toour response to Reviewer’s Comment: Comment #R1Were these the same participants? This study is from …

Reviewer’s Comment: Comment R7: Is this whole brain signal difference also present using frequency band analysis? Or is entropy the best tool for this task?

Authors’ Response: These are indeed interesting questions that deserve further clarification.

Although the use of entropy in its discrete form have been previously reported within the context of brain analysis (e.g., [1] and [2] Referencesbelow), DE (i.e., differential entropy or the entropy of a continuous random variable), appears to be first utilized in [3] (the Referencesbelow) in the context of studying the EEG frequency-domain. Following that study, Zhengand Lu [59] argued that since EEG data has the higher low frequency energy over high frequency energy, DE has the balance ability of discriminating EEG pattern between low and high frequency energy.”They further compared DE with such features as differential asymmetry (DASM), rational asymmetry (RASM), and power spectral density (PSD) in their study and showed that DE outperformed these features in the frequency-domain.

However, entropic measures have been mainly utilized (to the best of our knowledge) in time- than frequency-domain analyses and they are generally concerned with the temporal dynamical changes of the brain activity. In this respect, authors in [4] (the References below) have proven that the rate of change in DE approximates the neural spiking (i.e., also known as Fano factor [5], the Referencesbelow). It is also well-established that the brain optimizes its information processing through the entropy maximization ( References[6], chapter 4 and [7] below). Integral to such information-theoretic measures as entropy is the “Data Processing Inequality” (DPI) (in current version of the manuscript, [58], p. 34, Theorem 2.8.1) which states that the more the data manipulation (e.g., data pre/processing steps) the more the loss of information. This, in turn, helpsexplain why such measures are more useful when they applied on the time series than such transformations as the frequency that require further processing steps on data to extract different frequency bands (the potential detrimental effect of use of such dynamical system analysis tools for frequency-domain analysishas also been noted in the case of Granger causality ([8,9] (the References below))). However, it is important to note that other forms of entropic measures such as multiscale entropy (MSE, the References10 below) have been used to draw correspondence between the time- and frequency-domain analysis of the brain activity (the References[11-13] below).

We can include the above discussion in our manuscript if the reviewer considers they are necessary for understanding our study.

With regard to the second partof the comment (i.e., appropriateness of the use of entropy), in our manuscript (Section Introduction, lines 41-55) weprovided the motivation behindthe use of entropy and DE within the context of neural correlates of the affect that reads asfollows.

A common theme among these results is their focus on the change in brain activation to identify a specific [2629] or subset [30] of the brain regions as sources of different affect. More specifically, these results do not take into account the findings that indicate that the brain activation and its information content does not necessarily modulate [31] and that the stimuli with equivalent sensory and behavioural processing demands may not necessarily result in differential brain activation [32]. In this respect, a growing body of empirical [3335] and theoretical [36,37] findings provide compelling evidence for the crucial role of signal variability in the functioning of the brain. This variability that is thought to emerge from the interaction between individual neurons and neuronal circuits [38,39] occurs over broad spatiotemporal scales [40,41] and is hypothesized to reflect the cortical self-organized criticality [4245]: a state at which cortical information capacity is maximized [4648]. These observations, in turn, have motivated the viewpoints that identify the role of entropy in quantification of the variability in brain functioning [49] and cortical activity [50,51] from information processing capacity of working memory (WM) [52] and neural coding [53,54] to interplay between neural adaptation and behaviour [55], functional interactivity between the brain regions [56], and the state of consciousness [57].

    In light of these findings on the importance of brain variability, we take a different approach to analysis of the neural substrates of the affect, thereby considering the whole-brain variability as it is reflected in its entropy (hereafter differential entropy (DE) [58] i.e., entropy of a continuous random variable).

References

[11 Baddeley, R., Hancock, P. and Földiák, P. eds., 2008. Information theory and the brain. Cambridge University Press.

[2] Miller, G.A., 1956. The magical number seven, plus or minus two: Some limits on our capacity for processing information. Psychological review,63(2), p.81.

[3] R.-N. Duan, J.-Y. Zhu, and B.-L. Lu, “Differential entropy feature for EEG-based emotion classification,” in Proc. IEEE 6th Int. IEEE/EMBS Conf. Neural Eng. (NER), 2013, pp. 81–84.

[4] Keshmiri, S., Sumioka, H., Yamazaki, R. and Ishiguro, H., 2018. Differential Entropy Preserves Variational Information of Near-Infrared Spectroscopy Time Series Associated With Working Memory. Frontiers in neuroinformatics,12, p.33.

[5]Fano, U. (1947). Ionization yield of radiations. II. The fluctuations of the number of ions. Physical Review72, 26–29.

[6] Dayan, P. and Abbott, L.F., 2001.Theoretical neuroscience(Vol. 806). Cambridge, MA: MIT Press.

[7] Sharpee, T.O., Calhoun, A.J. and Chalasani, S.H., 2014. Information theory of adaptation in neurons, behavior, and mood. Current opinion in neurobiology,25, pp.47-53.

[8] Seth, A.K., 2010. A MATLAB toolbox for Granger causal connectivity analysis. Journal of neuroscience methods,186(2), pp.262-273.

[9] Seth, A.K., 2011. Granger causal connectivity analysis: a MATLAB toolbox. University of Sussex,40.

[10]Costa, M., Goldberger, A.L. and Peng, C.K., 2002. Multiscale entropy analysis of complex physiologic time series. Physical review letters,89(6), p.068102.

[11] Wang, D.J., Jann, K., Fan, C., Qiao, Y., Zang, Y.F., Lu, H. and Yang, Y., 2018. Neurophysiological basis of multi-scale entropy of brain complexity and its relationship with functional connectivity.Frontiers in neuroscience,12, p.352.

[12] Liu, M., Song, C., Liang, Y., Knöpfel, T. and Zhou, C., 2019. Assessing spatiotemporal variability of brain spontaneous activity by multiscale entropy and functional connectivity. NeuroImage,198, pp.198-220.

[13] McIntosh, A.R., Kovacevic, N. and Itier, R.J., 2008. Increased brain signal variability accompanies lower behavioral variability in development. PLoS computational biology,4(7), p.e1000106.

Round 3

Reviewer 3 Report

Thank you for your detailed responses to all the points raised in my previous review. I have no further comments.